

# Turbulent transport of energy across a forest and a semi-arid shrubland.

Tirtha Banerjee[1,3], Peter Brugger[1], Frederik De Roo[1], Konstantin Kröniger[1], Dan Yakir[2],
Eyal Rotenberg[2], and Matthias Mauder[1]

[1]Karlsruhe Institute of Technology (KIT), Institute of Meteorology and Climate Research, Atmospheric Environmental Research (IMK-IFU), 82467 Garmisch-Partenkirchen, Germany
[2]Department of Earth and Planetary Sciences (EPS), The Weizmann Institute of Science, Rehovot 76100, Israel
[3]Current affiliation: Earth and Environmental Sciences Division, Los Alamos National Laboratory, Los Alamos, New Mexico, USA

*Correspondence to:* Tirtha Banerjee (tirtha.banerjee@lanl.gov)

**Abstract.** The role of secondary circulations has recently been studied in the context of well defined surface heterogeneity in a semi-arid ecosystem where it was found that energy balance closure over a desert-forest system and the structure of the boundary layer was impacted by advection and flux divergence. As a part of the CliFF (Climate Feedbacks and benefits of semi-arid forests, a collaboration between KIT, Germany and the Weizmann Institute, Israel) campaign, we studied the boundary
layer dynamics and turbulent transport of energy corresponding to this effect in the Yatir forest situated in the Negev desert in Israel. The forest surrounded by small shrubs presents a distinct feature of surface heterogeneity, allowing us to study the differences between their interactions with the atmosphere above by conducting measurements with two EC stations and two Doppler LiDARs. As expected, the turbulence intensity and vertical fluxes of momentum and sensible heat are found to be higher above the forest compared to the shrubland. Turbulent statistics indicative of nonlocal motions are also found to differ
over the forest and shrubland and also display a strong diurnal cycle. The production of turbulent kinetic energy (TKE) over the forest is strongly mechanical, while buoyancy effects generate most of the TKE over the shrubland. Overall TKE production is much higher above the forest compared to the shrubland. The forest is also found to be more efficient in dissipating TKE. The TKE budget appears to be balanced on average both for the forest and shrubland, although the imbalance of the TKE budget, which contains the role of TKE transport, is found to be quite different in terms of their variation with atmospheric
stability and diurnal cycles for the forest and shrubland. The effect of very large mesoscale motions is also directly quantified following a recent formulation by Banerjee and Katul, 2013, using the measured longitudinal velocity variances and boundary layer heights. The difference of turbulent quantities and the relationships between the components of TKE budget are used to infer the characteristics of turbulent transport of energy between the desert and the forest.

## 1 Introduction

Understanding the interaction between vegetation canopies and atmosphere is a crucial component in quantification of biosphere-atmosphere exchange of heat, carbon dioxide, water and trace gas fluxes. It is also important for the development of numerical





weather and climate models where the fluxes in the canopy surface layer (CSL) and the atmospheric surface layer (ASL) are parameterized through bulk exchange coefficients of momentum and scalar. However, idealizations of the forest canopies as horizontally homogeneous momentum sinks and scalar sources introduces uncertainties in flux estimations and estimating diffusion coefficients. Presence of heterogeneities such as roughness transitions, complex topography, mesoscale circulations etc. are common sources of such uncertainties that give rise to nonlocal motions and secondary circulations. These secondary circulations can occur not only in forests but are generic characteristics of boundary layer flows over natural and man made landscapes with discongruity of land use types, surface moisture or temperature etc. (Higgins et al., 2013; Eder et al., 2015). Different types of land covers such as agricultural lands or urban areas can affect local energy balance closure and the structure of the overlying boundary layer as well as cloud formation and regional weather (Eder et al., 2015; Fuentes et al., 2016). Strong difference of surface properties and large swaths of such surface patches are known to induce secondary circulations (Mahfouf et al., 1987; Dalu and Pielke, 1993; Raupach and Finnigan, 1995; Courault et al., 2007; van Heerwaarden and Guerau de Arellano, 2008; Garcia-Carreras et al., 2010; Banerjee et al., 2013; Dixon et al., 2013; Sühring and Raasch, 2013; Kang and Lenschow, 2014; Van Heerwaarden et al., 2014). Recent works by Mauder et al. (2007), Stoy et al. (2013) and Eder et al. (2014) have suggested that non-closure of energy balance is also related to advection and flux divergence due to secondary circulations (Kanda et al., 2004; Foken, 2008). The non-closure of the energy balance refers to the fact that the available energy $R_n - G$ is often higher than the turbulent energy $H + LE$ in micrometeorological sites, where $R_n$ is net radiation, $G$ is soil heat flux, $H$ is sensible heat flux and $LE$ is latent heat flux. Thus it is established that studies involving surface heterogeneities such as difference of roughness characteristics and albedo are crucial for the advancements of our understanding into biosphere-atmosphere interaction since the quasi-universal scaling laws of turbulent moments and simple parametrizations of exchange coefficients are disturbed and rendered non-operational.

Several studies have attempted to study the nature of turbulence across a roughness transition such as a grassland and a forest canopy by means of experimental and numerical methods (Li et al., 1990; Peltola, 1996; Irvine et al., 1997; Belcher et al., 2003; Yang et al., 2006; Cassiani et al., 2008; Detto et al., 2008; Dupont and Brunet, 2009; Dalpe and Masson, 2009; Fesquet et al., 2009; Gavrilov et al., 2010, 2011; Huang et al., 2011; Rominger and Nepf, 2011; Schlegel et al., 2012; Banerjee et al., 2013; Chatziefstratiou et al., 2014; Markfort et al., 2014; Kanani-Sühring and Raasch, 2015; Queck et al., 2016; Kroeniger et al., 2017) and documented several length scales associated with the roughness transitions, recirculation zones and as well as the nature of the turbulent momentum budget. However, all of these studies are concerned with the flow adjustment in the immediate vicinity of the roughness transition (edges or gaps). Eder et al. (2015) have studied the dynamics of the convective boundary layer over a well defined surface heterogeneity- namely the Yatir forest and the shrubland surrounding it which are located in the norther part of the Negev desert in Israel. Eddy covariance (EC) and Doppler LiDAR measurements were conducted by Eder et al. (2015) in two sites, one in the forest and one in the desert approximately 6.5 km apart. The forest has a darker surface and consequently lower albedo (12.5 %) than the desert (33.7 %). Moreover, the higher surface roughness of the forest results in higher turbulence intensity, which leads to more efficient heat transfer above the forest, a phenomenon called canopy convector effect (Rotenberg and Yakir, 2011; Banerjee et al., 2017). The region being very dry, there is very little latent heat flux (Bowen ratio > 10 over the summer), resulting in spatial difference of surface buoyancy flux of 220-290 $\mathrm{Wm}^{-2}$





between the desert and forest. Furthermore, the length scale of surface heterogeneities (6-10 km) is larger than the minimal length scale needed for development of secondary circulations $L_{rau} = C_{Rau}U/w_* \approx 2-5$ km (Raupach and Finnigan, 1995; Eder et al., 2015), where $U$ is mean wind speed, $w_*$ is the convective velocity scale and $C_{Rau} = 0.8$, an empirical parameter, so that it is possible for secondary circulations to develop.

The present work is an attempt to examine this hypothesis of secondary circulations in more detail. We use eddy covariance and Doppler LiDAR measurements at two sites over the shrubland and the Yatir forest 4.3 km apart, where the shrubland is upwind of the forest in the path of the principal wind direction (there exists a heat induced low pressure system to the east, resulting in the main wind direction from the north west). We investigate the individual components of the turbulent kinetic energy budget, as well as the nature of advection and turbulent transport over the forest and desert and determine if there is a

relationship between them. The role of large scale structures is also investigated.

## 2   Method

### 2.1   Theory

The turbulent kinetic energy (TKE) budget without invoking any special assumption is given by (Stull, 2012)

$$\frac{\partial e}{\partial t} + U_j \frac{\partial e}{\partial x_j} = \delta_{i3}\frac{g}{T}\left(\overline{u_i'T'}\right) - \overline{u_i'u_j'}\frac{\partial U_i}{\partial x_j} - \frac{\partial\left(\overline{u_j'e}\right)}{\partial x_j} - \frac{1}{\rho}\frac{\partial\left(\overline{u_i'p'}\right)}{\partial x_i} - \epsilon, \qquad (1)$$

where $i$ and $j$ are usual tensor indices which can take the values of 1, 2 and 3, to indicate $x$, $y$ and $z$ directions respectively and $\delta_{i3}$ is the Kronecker delta. $e = (1/2)(\sigma_u^2 + \sigma_v^2 + \sigma_w^2) = (1/2)(\overline{u'^2} + \overline{v'^2} + \overline{w'^2})$ is the TKE, $U$ denotes mean longitudinal velocity, $u'$, $v'$ and $w'$ denote the fluctuations from mean for the longitudinal, transverse and vertical velocity components, $g$ is acceleration due to gravity, $T$ denotes mean potential temperature, $T'$ is the potential temperature fluctuation, $p'$ is the dynamic pressure perturbation, $\rho$ is density of air. The first term on the left hand side (LHS) denotes storage or TKE tendency. The

second term on the LHS indicates advection of TKE by mean wind flow. The first term on the right hand side (RHS) denotes buoyant production/destruction of TKE. The second term on the RHS denotes mechanical/shear production of TKE. The third term on RHS denotes turbulent transport of TKE and can also be called turbulent flux divergence. The fourth term on RHS denotes transport of TKE by pressure velocity correlation. $\epsilon$ is the dissipation of TKE.

    Expanding the equations in terms of $x$, $y$ and $z$ coordinates, the full TKE budget can be written as equation A1 as shown in

appendix A. Since it is difficult to keep track of the full equation due to the sheer number of terms, it would be easier to use a simple form of the TKE budget (Stull, 2012)

$$0 = -\overline{u'w'}\frac{dU}{dz} + \frac{g}{T}\overline{w'T'} - \epsilon - Imbalance. \qquad (2)$$

where the $Imbalance$ is defined in equation A2. Note that $\overline{u'w'}$ and $\overline{w'T'}$ denote vertical momentum flux and sensible heat flux respectively. Also notice that if the term $Imbalance$ is set to zero, one recovers the TKE budget for an idealized surface

layer where the coordinate system is aligned with the mean wind, and a planar, homogeneous flow with zero subsidence is





assumed. Since our objective in the current problem is to study the effect of heterogeneity, we cannot make these assumptions. Moreover, we are also constrained by being able to measure only at two single points in space quite far apart. Single point eddy covariance measurements cannot compute spatial gradients, and the pressure perturbations are not measured either. Thus explicit computations of the imbalance terms are not possible. Due to the three dimensional nature of the problem, it is also

difficult to anticipate what degrees of assumptions are sufficient so that some of the terms can be ignored safely.

Under these constraints, a strategy is needed to evaluate the TKE budget. As will be shown later, we are able to compute the terms in the equation 2 except $Imbalance$ prognostically. Thus the dominant mechanical production term, the buoyant production/destruction term and the dissipation term will be evaluated directly from the data. The residual of the budget will be described as the imbalance as per equation 3 which would diagnostically contain the effects of advection and transport terms.

The advantage of using this strategy is that since the original TKE budget equation has to be closed, the errors in computing the production and dissipation terms can also be assumed to be inside the $Imbalance$ term.

$$Imbalance = -\overline{u'w'}\frac{dU}{dz} + \frac{g}{T}\overline{w'T'} - \epsilon. \tag{3}$$

To compute the mechanical production term, we momentarily assume that the TKE budget is well balanced and Monin Obukhov Stability Theory (MOST) (Monin and Obukhov, 1954) is valid (Banerjee et al., 2016). This allows us to write

$$\frac{dU}{dz} = \phi_m(\zeta)\frac{u_*}{\kappa z}, \tag{4}$$

where $\phi_m$ is the stability correction function for momentum which varies with the stability parameter $\zeta = z/L$ and $\kappa = 0.4$, the von-Kármán constant. $u_* = \sqrt{|\overline{u'w'}|}$ is the friction velocity, $z$ is the measurement height and $L = -u_*^3/(\kappa(g/T)\overline{w'T'})$ is the Obukhov length. The standard MOST scaling relations for $\phi_m$ are used, i.e., $\phi_m = 0.74 + 4.7\zeta$ for stable ($\zeta > 0$) and $\phi_m = (1 - 16\zeta)^{-1/4}$ for unstable ($\zeta < 0$) stratification (Businger et al., 1971; Dyer, 1974).

Equation 4 allows us to compute the mechanical production term in equation 2 as $-\overline{u'w'}dU/dz = \phi_m u_*^3/(\kappa z)$. The buoyancy term can directly be computed from the EC measurements as well. To compute the dissipation term $\epsilon$, we use the scaling relation of second order structure function $D_{uu} = \overline{[u(x+r) - u(x)]^2}$ in the inertial subrange (Salesky et al., 2013; Banerjee et al., 2015, 2016; Li et al., 2016)

$$D_{uu}(r) = C_u\epsilon^{2/3}r^{2/3}, \tag{5}$$

where $C_u \approx 2$ (Stull, 2012), and $r$ is the spatial lag in the longitudinal direction which can be computed by multiplying the sampling time interval with the mean longitudinal velocity, assuming that Taylor's frozen turbulence hypothesis is valid ($r = |u|\Delta t$). The range of $r$ where this relation is valid is found to be between 0.2 to 2 m and $\epsilon$ is found by regression of equation 5. Note that the computation of $\epsilon$ is independent from any assumptions used to compute the production terms.

## 2.2 Research site

The measurements were conducted in the Yatir forest and the surrounding shrubland in Israel between 18th August and 30th August, 2015 as part of the "Climate feedbacks and benefits of semi-arid forests" (CliFF) campaign, a joint collaboration





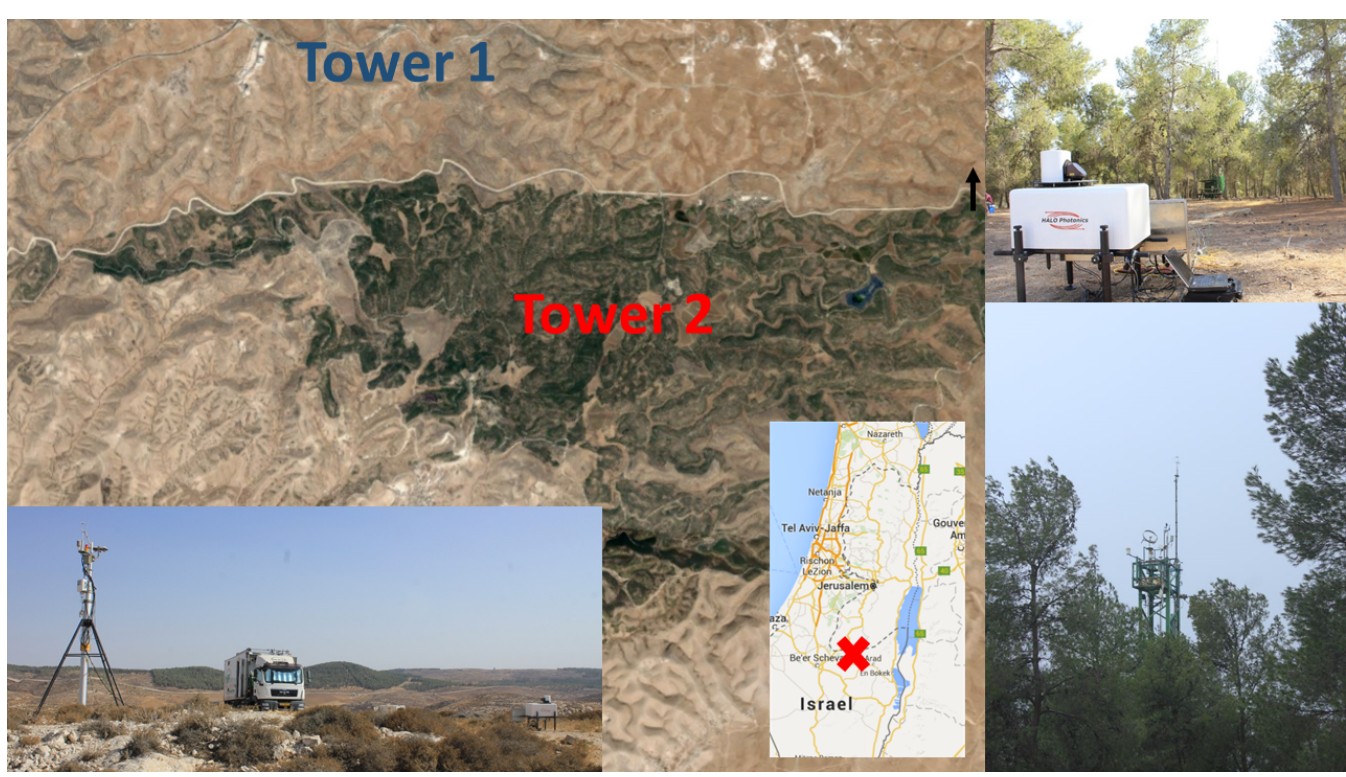

**Figure 1.** Map of Yatir forest in Israel and locations of the measurement stations. Insets: snapshots of measurement set-ups.





**Table 1.** Instrument specification and settings of the Doppler LiDARs. From top to bottom: Serial number of the forest and desert LiDAR, pulse length of the laser pulse at full width at half maximum, range gate length, pulse repetition frequency, number of averaged pulses for a backscatter coefficient profile and the wavelength of the emitted laser pulse (short wavelength infrared).

| Serial numbers | 0114-74 and 0114-75 |
| --- | --- |
| Pulse length | 60 m |
| Range gate length | 18 m |
| Pulse repetition frequency | 15 kHz |
| Averaged pulses per estimate | 15000 |
| Wavelength of laser light | 1.5 $\mu$m |

between Karlsruhe Institute of Technology (KIT), Germany and the Weizmann Institute, Israel. Fig 1 gives an idea about the locations of the EC towers. Tower 1 (Latitude 31.375728, Longitude 35.024262) was located at the semi-arid shrubland 620 m above sea level and tower 2 (Latitude 31.345315, Longitude 35.052224) was located inside the forest 660 m above sea level. The linear distance between the two locations was measured to be 4.3 km and as can be observed from figure 1, there

is a distinct surface heterogeneity between the two sites. The climate of the area is in between Mediterranean and semi-arid, with a mean annual precipitation of about 285 mm (Eder et al., 2015). The trees in the forest were mostly Aleppo Pine (*Pinus halepensis*), average 10 m in height with negligible height variation. The surrounding land was sparsely populated by small shrubs and in the dry season where the measurements were conducted, was mostly free of vegetation. Thus it is referred to as 'desert' for easy distinction (Eder et al., 2015). The measurement height for the forest was 19 m above ground (9 m

above the canopy height). A mobile mast was used over the desert and the measurement height was 9 m until 23rd August, after which it was changed to 15 m for the remaining period. However, the raising of the mast should not have affected the measurement of turbulent fluxes since it was done within the constant flux layer. In this zone of the atmospheric surface layer, the longitudinal and crosswise velocity variances decrease logarithmically with height and the vertical velocity variance shows an independence with height (Townsend, 1976; Perry and Chong, 1982; Marusic et al., 2013; Banerjee and Katul, 2013a).

Thus there is no evidence to support that increasing of measurement height affected turbulence measurements. High frequency turbulent data were collected at 20 Hz and 30 minutes averaging periods were used for both sites. In addition, two Doppler LiDARs were used at the two locations to measure the boundary layer height as well.

In addition, two Doppler LiDARs were used at the two locations to measure the boundary layer height as well. The Doppler LiDARs used were StreamLine systems from HaloPhotonics. They were operated in a vertical stare mode most of the time

(interrupted every half hour for less than 90 seconds). Technical specifications and instrument settings of the Doppler LiDARs are given in Table 1. For retrieval of the boundary layer height the three largest negative backscatter gradients were computed from the backscatter profiles of a 10 minutes interval to estimate aerosol layer heights. Then a modified algorithm of Lotteraner and Piringer (2016) was used for post-processing of the aerosol layer time series to get a time series of the boundary layer height. The Doppler LiDAR at tower 1 was not working from 19.08.2015 - 15:00 UTC until 21.08.2015 - 10:30 UTC and very

shortly on the 23.08.2015 around 10:00 UTC due to power cuts.



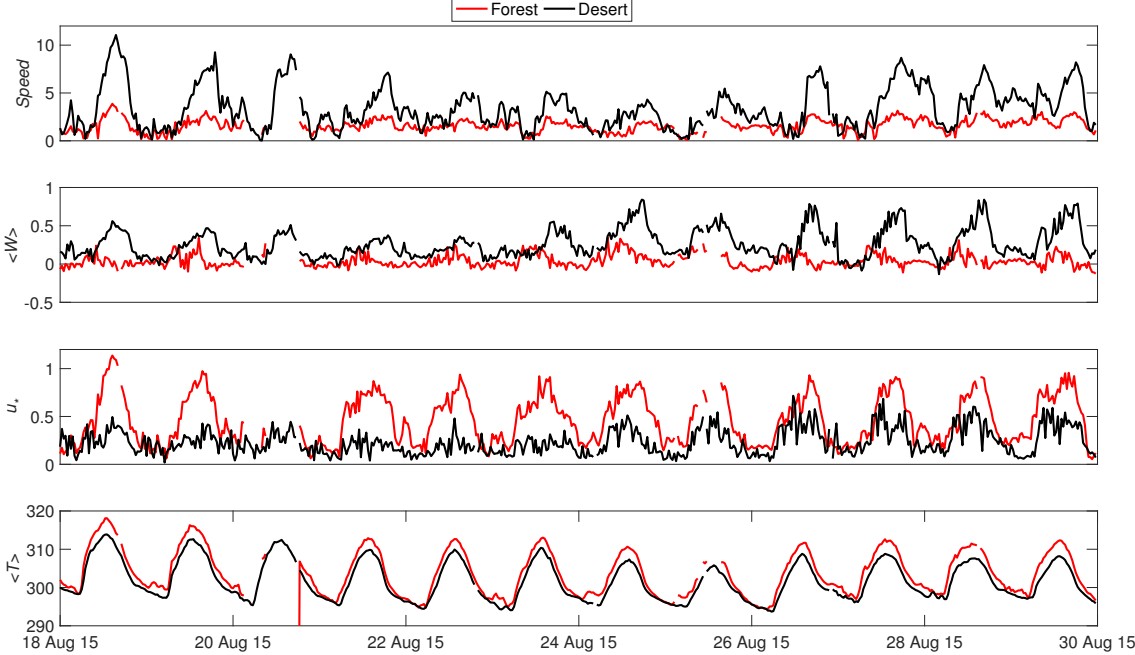

**Figure 2.** Time series of mean speed ($\mathrm{ms^{-1}}$), mean vertical velocity ($\mathrm{ms^{-1}}$), friction velocity ($\mathrm{ms^{-1}}$) and mean potential temperature (K) for the measurement period. Black line indicates desert and red line indicates forest.

## 3   Results and Discussion

### 3.1   Time series of turbulence statistics

Time series of mean speed ($\mathrm{ms^{-1}}$), mean vertical velocity ($W$, $\mathrm{ms^{-1}}$) friction velocity ($u_*$, $\mathrm{ms^{-1}}$), mean near surface air (potential) temperature ($T$, K), for the measurement period are shown in figure 2. Figure 3 shows time series of longitudinal

5   velocity variance ($\overline{u'u'}$, $\mathrm{m^2s^{-2}}$), vertical velocity variance ($\overline{w'w'}$, $\mathrm{m^2s^{-2}}$), momentum flux ($\overline{u'w'}$, $\mathrm{m^2s^{-2}}$) and sensible heat flux ($\overline{w'T'}$, $\mathrm{Kms^{-1}}$). Thicker line indicates desert and thinner line indicates forest. As noticed, the desert is associated with a higher wind speed and higher mean vertical velocity. Although the friction velocity ($u_*$) over the forest is much higher compared to the desert, especially in the daytime, which is expected because of higher surface roughness over the forest. $u_*$ above both the forest and desert shows a strong diurnal cycle. However, there seems to be a prominent increase of $u_*$ over the desert after

10   24th August. This can be attributed to mesoscale motions appearing over the region, which will be discussed later. The higher mean speeds over the desert indicate of existence of large energetic eddies, possibly associated with secondary circulations. There could be several factors responsible for the secondary circulations. The gentle topography around the desert could result in the strong vertical updrafts above the desert. The consistent presence of the sea breeze from the north west due to the heat



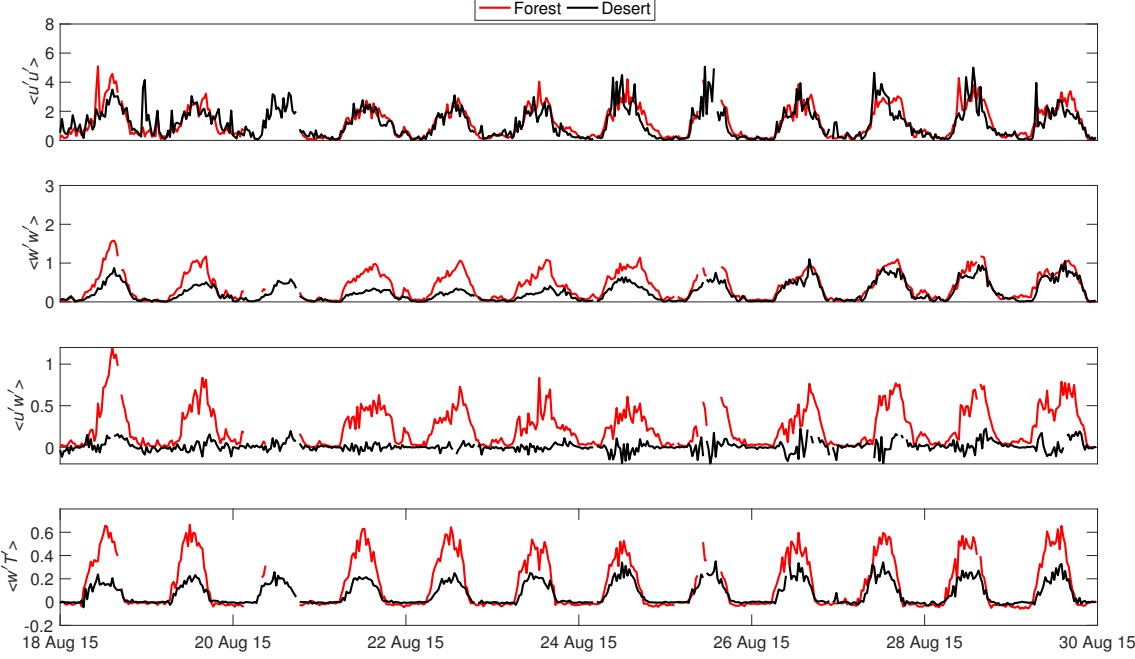

**Figure 3.** Time series of longitudinal velocity variance ($\mathrm{m^2 s^{-2}}$), vertical velocity variance ($\mathrm{m^2 s^{-2}}$), momentum flux ($\mathrm{m^2 s^{-2}}$) and sensible heat flux ($\mathrm{Kms^{-1}}$) for the measurement period. Black line indicates desert and red line indicates forest.

induced low pressure system to the west could be another major feature of the secondary circulations. However, the secondary circulations cannot raise the turbulence levels above the desert since mechanical forcing by roughness is absent. Interestingly, the near surface air temperatures over both the forest and desert show a strong diurnal cycle but their daytime differences are not very high. The longitudinal velocity variance $\overline{u'u'}$ over the forest and desert show similar variations over time, and the effects

5   of a large scale energetic system are also visible in $\overline{u'u'}$ for both the desert and forest. The vertical velocity variance $\overline{w'w'}$ over the forest is higher than its desert counterpart, however, after 24th August, the levels of $\overline{w'w'}$ over desert increases as well and become similar to the forest. Thus this large scale structure influences the already existing secondary circulation, and it ramps up turbulence levels in both velocity components differently over the desert and the forest. The vertical momentum flux $\overline{u'w'}$ over the forest is much higher compared to the desert-which is also expected because of the higher surface roughness of the

10   forest, making it a much more efficient momentum sink compared to the desert. Note that the shear transport of momentum flux is still much more effective over the forest compared to the desert because of roughness effects even though the mean quantities can be higher over the desert. The sensible heat flux $\overline{w'T'}$ over the forest is also higher as discussed before due to the canopy convector effect, but the mesoscale structure increases the sensible heat flux above the desert after 24th August.




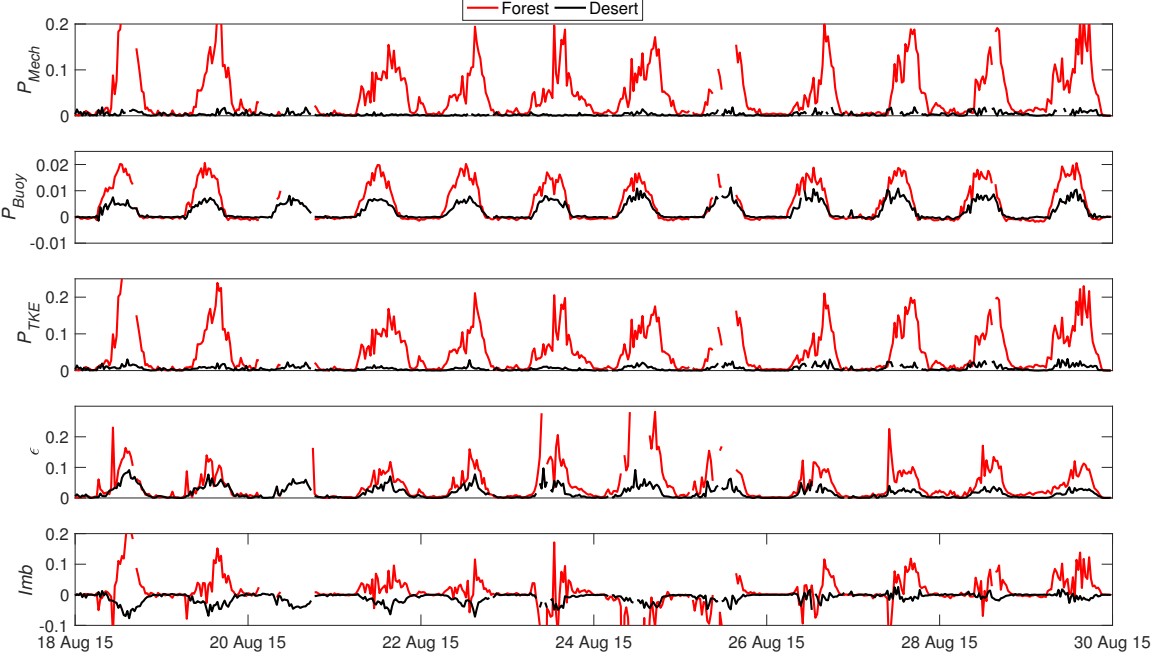

**Figure 4.** Time series of mechanical production of TKE ($\mathrm{m^2s^{-3}}$), buoyant production of TKE ($\mathrm{m^2s^{-3}}$), full TKE production ($\mathrm{m^2s^{-3}}$), dissipation of TKE ($\mathrm{m^2s^{-3}}$) and imbalance of TKE ($\mathrm{m^2s^{-3}}$). Black line indicates desert and red line indicates forest.

## 3.2 Nature of TKE budget

Figure 4 shows the time series of the components of the TKE budget as discussed in section 2.1. The first row shows mechanical production of TKE ($P_{Mech}$, $\mathrm{m^2s^{-3}}$), the second row shows buoyant production of TKE ($P_{Buoy}$, $\mathrm{m^2s^{-3}}$), the third row shows full TKE production ($P_{TKE}$, $\mathrm{m^2s^{-3}}$), the fourth row shows dissipation of TKE ($\epsilon$, $\mathrm{m^2s^{-3}}$) and the fifth row shows imbalance

5   of TKE ($Imb$, $\mathrm{m^2s^{-3}}$). Thicker line indicates desert and thinner line indicates forest. As noticed in figure 4, the production of turbulence is mostly by mechanical or shear forcing because of the roughness of the forest, whereas mechanical production of TKE over desert is very small and does not also have a strong diurnal cycle like the forest, although it increases slightly after 24th August. On the other hand, TKE production over the desert is mostly carried by buoyancy. Buoyant TKE production over the forest is slightly larger over the forest but of similar order of magnitude as the desert. The buoyant TKE production

10  over the desert is also higher after 24th August. Given the moderate temperature difference between the desert and the forest, the difference of their corresponding buoyant TKE production is interesting. It also indicates that mechanical forcing, and not buoyancy makes a huge difference in the turbulence generation over the desert and the forest. The diurnal cycle of the TKE dissipation $\epsilon$ is interesting as well. The dissipation of TKE seems to be higher above the forest as well compared to the desert. Although the effect of the large scale structure after 24th August seems to dampen the $\epsilon$ over the desert while its





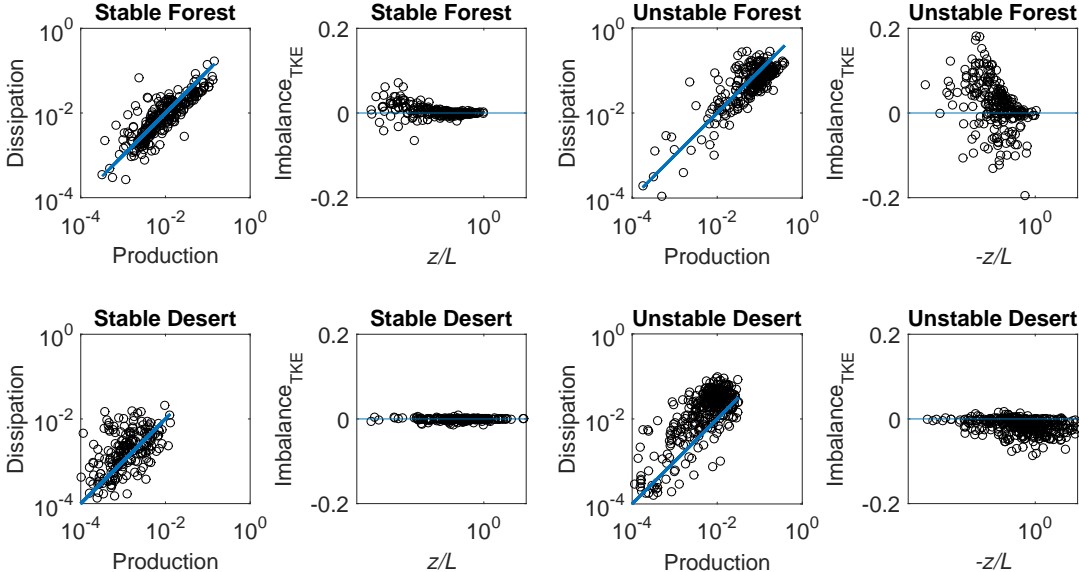

**Figure 5.** First row: forest, second row: desert. First column: TKE production vs dissipation for stable conditions. Second column: variation of TKE imbalance with stability parameter for stable conditions. Third column: TKE production vs dissipation for unstable conditions. Fourth column: variation of TKE imbalance with stability parameter for unstable conditions.

effects on the $\epsilon$ over the forest are not very conspicuous. The diurnal cycles of the TKE imbalance computed by equation 3 is also very interesting. The imbalance over the forest is often positive over the daytime, while on the desert it is often negative, highlighting the difference of turbulent transport and advection over the two different regimes. Also notice that the positive imbalance for forest and negative imbalance for desert almost have a phase (anti)synchronization, indicting that the turbulence

5    above forest and desert are responsive to one another and they are part of a coupled system, indicating again towards the role of the secondary circulations.

Figure 5 highlights the nature of TKE budget over the forest and the desert in more details. The first row shows TKE budgets for the forest and the second row shows the same for the desert. The first column shows TKE production vs dissipation for stable conditions. The second column shows variation of TKE imbalance with the stability parameter $\zeta = z/L$ for stable

10   conditions ($\zeta > 0$). The third column shows TKE production vs dissipation for unstable conditions and the fourth column shows the variation of TKE imbalance with stability parameter for unstable conditions ($\zeta < 0$). As observed, the Range of TKE production and dissipation are higher for unstable conditions for both the forest and desert compared to stable conditions. On average, the TKE budget is closed as seen from the one to one solid line for both the desert and forest, although there is more uncertainty in the TKE closure for the desert compared to the forest for both stable and unstable conditions. However, the

15   TKE imbalance for the forest increases with near neutral conditions and reduces as the stratification becomes more convective or highly stable. On the other hand, the TKE imbalance for desert does not change much with stability for the stable conditions





and actually increases more with more convective conditions, which is a directly opposite to the nature of TKE imbalance over the forest. This behavior gives us more insight into the nature of turbulent transport over the forest and desert and highlights their difference. More convective conditions means more TKE transport by means of advection, subsidence and flux divergence over the desert.

### 3.3 Transport of TKE over desert and forest

Figure 6 shows the interrelationship between the nature of turbulent transport over the desert and forest. Panel (a) depicts the TKE imbalance over desert vs the net production of TKE over the forest. As observed, there is a significant correlation (0.5) between them, indicating that the advection and transport of TKE by flux divergence and pressure fluctuations reach downstream by means of the secondary circulations and produces TKE over the forest. On the other hand, the converse is not true, as observed in panel (b) of figure 6. There is little correlation between the Imbalance of TKE over forest and the production of TKE over desert (0.14). As observed in panel (c), the production over desert is also well correlated with the production over forest (0.3) as both the desert and forest are subjected to the same forcing. However the TKE production over desert is not that well correlated with the TKE imbalance over the desert as seen in panel (d). Thus while there should be some cross correlation in panel (a) because of desert production, that is not the only effect. The nonlocal large scale motions contribute to the transport over desert (without significantly altering TKE production over desert) which in turn cause TKE production above the forest because of the higher mechanical forcing.

Thus it can be stated that the effects of secondary circulations are transported from over the desert towards the forest following the background wind direction, and not the other way around. It is worth noting here that the term 'secondary circulation' has been used somewhat loosely here and contain the effects of horizontal transport as well, since partitioning the imbalance term is not possible within the scope of this campaign. In the case of transport from the forest towards the desert, it is more likely that horizontal advection is the main mechanism.

### 3.4 Effect of nonlocal motions

Figure 7 shows the time series of the triple moments $\overline{w'w'u'}$, $\overline{w'w'w'}$ and $\overline{w'w'T'}$ in the first three rows. The vertical velocity skewness term $\overline{w'w'w'}$ (2nd row) is of importance as it appears in the transport term of the TKE budget (equation 2) and is a measure of non-Gaussian turbulence, which indicates the presence of non-local coherent motions such as sweeps and ejections. Note that the vertical velocity skewness is often negative above the canopy which is consistent with the generic feature of canopy turbulence (Kaimal and Finnigan, 1994; Chamecki, 2013; Dias-Junior et al., 2015). What is perhaps more interesting is that the daytime vertical velocity skewness over the desert is often positive, indicating again of the presence of nonlocal coherent structures active over the desert. The measure of skewness increases over both the forest and the desert after 24th August, indicating the arrival of the large scale mesoscale system. The other two terms $\overline{w'w'u'}$ and $\overline{w'w'T'}$ are also associated with turbulent transport of momentum and heat by as evident from their respective budget equations (Raupach et al.,





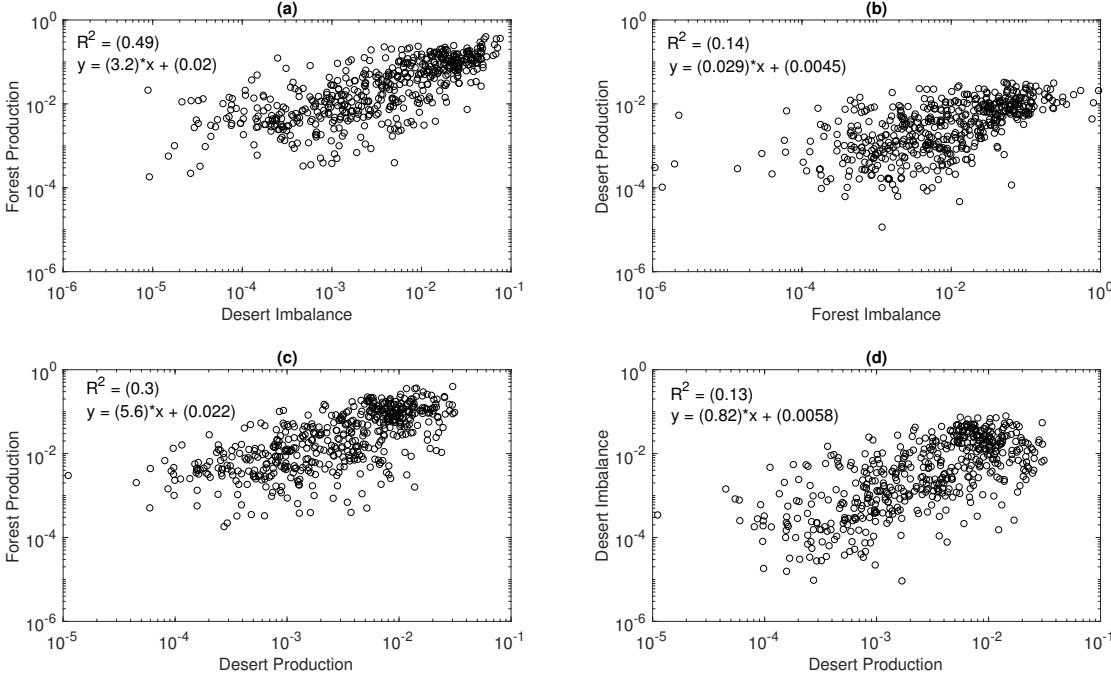

**Figure 6.** (a) TKE imbalance for desert vs TKE production for forest. (b) TKE imbalance for forest vs TKE production for desert. (c) TKE production for desert vs TKE production for forest. (d) TKE production for desert vs TKE imbalance for desert.

1986; Zhuang and Amiro, 1994; Cava et al., 2006; Katul et al., 2013; Banerjee et al., 2017a).

$$\frac{\partial \overline{w'u'}}{\partial t} = 0 = -\overline{w'w'}\frac{\partial \overline{U}}{\partial z} - \frac{\partial \left(\overline{w'w'u'}\right)}{\partial z} + \frac{g}{\overline{T}}\overline{u'T'} - \frac{1}{\rho}\left(\overline{u'\frac{\partial p'}{\partial z}}\right); \tag{6}$$

and

$$\frac{\partial \overline{w'T'}}{\partial t} = 0 = -\overline{w'w'}\frac{\partial \overline{T}}{\partial z} - \frac{\partial \left(\overline{w'w'T'}\right)}{\partial z} + \frac{g}{\overline{T}}\overline{T'T'} - \frac{1}{\rho}\left(\overline{T'\frac{\partial p'}{\partial z}}\right). \tag{7}$$

5 Moreover, the triple moments have been shown to be directly correlated with the relative contributions of nonlocal events such as sweeps and ejections as (Nakagawa and Nezu, 1977; Raupach et al., 1986; Cava et al., 2006; Katul et al., 2013; Banerjee et al., 2017a). Notice that momentum transport term $\overline{w'w'u'}$ is also opposite in nature for the desert and forest and it shows a strong diurnal cycle. The mesoscale structure after 24th August increases this momentum transport for both the forest and desert. However, the diurnal cycle of the heat transport term $\overline{w'w'T'}$ is not as strong as its momentum counterpart, but it is

10 often found to be larger over the desert compared to the forest, consistent with the findings from the TKE budget that shows heat is transported from over the desert towards the forest. The fourth and fifth rows of figure 7 show the timeseries of integral timescale of horizontal ($In_u$) and vertical ($In_w$) velocity components in seconds. $In_u$ and $In_w$ for every half hour time period are computed by integrating the the normalized autocorrelation function of $u$ and $w$ until the first zero crossing (Kaimal and



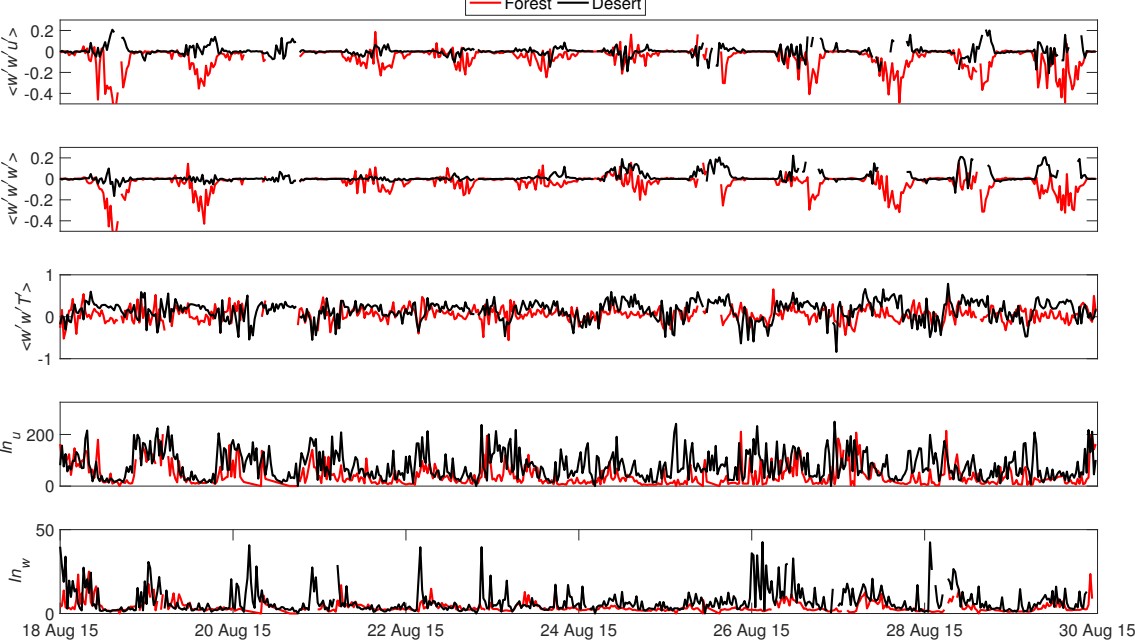

**Figure 7.** Top three panels: time series of triple moments $\overline{w'w'u'}$, $\overline{w'w'w'}$ and $\overline{w'w'T'}$ ($\mathrm{m^3 s^{-3}}$). Bottom two panels show the integral time scales of horizontal ($In_u$) and vertical velocities ($In_w$) in seconds.

Finnigan, 1994). They can be interpreted as the characteristic time scale of the most energetic eddies in each direction. As noticed in figure 7, time scales in the horizontal directions are larger compared to the vertical direction. More interesting is the observation that the integral time scales for the eddies above the desert are larger than the forest- both of which increase after 24th. However, this is another indicator of the transport by secondary circulations above the desert.

## 3.5 Effect of very large scale motions (VLSM)

Finally, the effects of the mesoscale structure are further analyzed in figure 8. The top panel shows the boundary layer heights ($\delta$) over the desert and forest estimated from the backscatter profiles of the Doppler LiDARs. The difference of $\delta$ between the desert and forest are not very large although the $\delta$ over forest is generally higher than the desert because of the higher levels of turbulence above it. $\delta$ increases over both the forest and desert after 24th August when Israel is affected by wave activity in the westerlies. We seek to quantify the signature of such a mesoscale system on the turbulence above the desert-forest system and a novel technique is used. Banerjee and Katul (2013a) developed a theoretical framework based on spectral theory to describe the scaling law of the longitudinal velocity variance $\sigma_u^2/u_*^2$ in the surface layer. This scaling law includes a bulk parameter $\alpha$





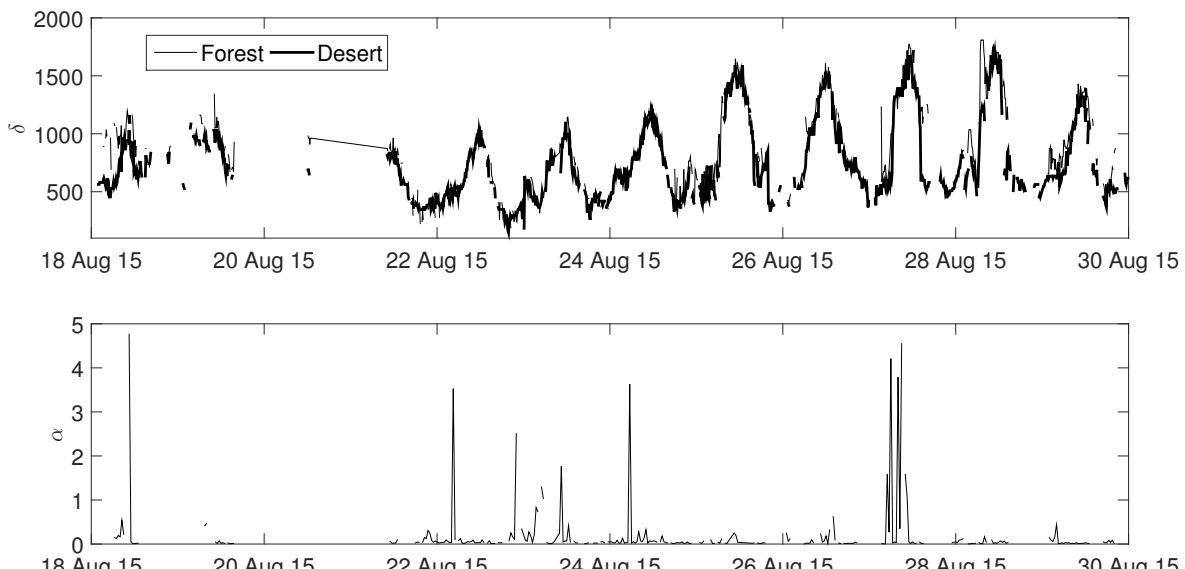

**Figure 8.** Top panel: Boundary layer heights ($\delta$ in m) measured using the Doppler LiDARs. Bottom panel: bulk measure of very large scale motion above the forest.

that represents the bulk effects of very large scale motions (VLSM) as well as the boundary layer height $\delta$:

$$\frac{\sigma_u^2}{u_*^2} = B_1 - A_1 ln\left(\frac{z}{\delta}\right),$$ (8)

where

$$B_1 = C'_{TKE}(1 + ln(\alpha)) + \frac{3}{2}\frac{C''_K}{\kappa^{2/3}}$$ (9)

and

$$A_1 = C'_{TKE};$$ (10)

where $C''_K = 0.55$, $\kappa = 0.4$ (von Kármán constant) and $C'_{TKE} = 1.0$. A value of $\alpha > 1$ indicates the presence of VLSM since $ln(1) = 0$. $\alpha$ is computed using equations 8, 9 and 10 with the experimental data over the forest and plotted in the second panel of figure 8. The data over the desert is found to be ill conditioned to compute $\alpha$, however since the synoptic conditions over both the desert and forest are similar as observed from the boundary layer heights, it can be taken as representative for both. As observed, there are a number of large peaks of $\alpha > 1$ after 24th August which confirms the presence of VLSM and supports the interpretations of previous findings in this manuscript. It is worth mentioning here that the exact nature/type of the very large scale structures can not be discerned here - i.e., it cannot be pinpointed at a sea-breeze or other mesoscale events. However, a higher value of $\alpha$ would indicate a stronger large scale motion, the effect of which is to include more large scale low frequency motions in the turbulent spectra (Banerjee and Katul, 2013a).



## 4  Conclusions

We studied the nature of turbulent transport over a well defined surface heterogeneity comprising of a desert and forest in the Yatir semi-arid area in Israel. Eddy covariance and Doppler LiDAR measurements were conducted for 12 days between 18th and 31st August, 2015 over two locations in the forest and the shrubland (referred to as 'desert' for the almost complete lack of

vegetation during the observation period). Earlier campaigns in this area focused on energy balance closure and hypothesized about the existence of secondary circulations because of surface heterogeneity. The present work was aimed to study the nature of turbulent transport over the forest and the desert in more detail to address the following questions:

1. How does the Yatir forest affect the boundary layer dynamics such as eddy size distribution, boundary layer height and diurnal variations of turbulent statistics and fluxes compared to the surrounding desert?

2. Can the existence of secondary circulation be confirmed?

3. Is there any horizontal energy transport between the forest and the desert and how does it vary with time?

4. What is the effect of mesoscale motions on the turbulent dynamics?

To answer the above mentioned questions, we computed half hour average turbulent statistics for both the desert and forest and looked at their diurnal variations. We also computed individual components of the turbulent kinetic energy (TKE) budget

and argued that the turbulent transport of energy should be contained in the imbalance of the TKE budget, which consists of the effects of advection, transport by turbulent flux divergence and pressure velocity interactions, since we could not compute those terms explicitly. Moreover, we also computed triple moments which are associated with nonlocal motions and coherent structures and integral time scales, which are associated with the most energetic eddies. We used the measured boundary layer heights to compute a first order bulk parameter which can directly quantify the presence of very large scale motions. The

findings to the questions are listed below:

1. The forest is found to be associated with a higher level of turbulent intensity because of higher roughness although the desert reported higher mean speeds and vertical updrafts possibly due to the presence of secondary circulations. Gentle topography around the desert might contribute to the updrafts over the desert as well. The smaller roughness of the desert is also responsible for higher wind speeds above the desert. There is little air temperature difference between the desert

and the forest, although the mean velocities and temperature have strong diurnal cycles. Momentum and heat flux are also found to be stronger above the forest. The presence of a large scale system after 24th August seem to strengthen the secondary circulation above the desert and enhance the turbulent fluxes as well as the turbulent intensity above the desert.

2. The role of secondary circulations can be better understood once the components of the TKE budget are studied. Over the

forest the production of turbulence is mechanical, while over the desert, TKE production is mostly carried by buoyancy. The forest is more efficient in dissipating TKE as well. The imbalance of TKE is taken as the indicator of TKE transport





and is found to vary diurnally almost anti synchronously over the desert and forest - confirming the role of a secondary circulation. The TKE budget is closed better over the forest compared to the desert. This imbalance and consequently the secondary circulation also varies differently with atmospheric stability. For the desert, it is higher for more stable conditions, possibly indicating of a secondary circulation driven by advection under a temperature inversion. Turbulent triple moments which are indicators of nonlocal motions and coherent structures also show strong variability over the desert and opposite in signs, also confirming the role of secondary circulations. The integral time scales are found to be higher over the desert compared to the forest. This suggests that the secondary circulations that transport energy are more active over the desert- however, they cannot produce much turbulence over the desert since they only rely on buoyancy driven turbulence as mechanical forcing is missing over the desert. This is also highlighted by the fact that mean velocities are higher above the desert while turbulent fluctuations are higher above the forest.

3. To elucidate the role of horizontal transport between the desert and the forest, we studied the correlation between the TKE imbalance over the desert and the TKE production over the forest. The moderately high correlation suggests that the secondary circulation is transported from over the desert towards the forest, enhancing TKE production over the forest. The low correlation between the TKE imbalance over the forest and TKE production over the desert confirms the directionality of this horizontal exchange, which is from the desert towards the forest and not the other way around.

4. Weather data confirms that Rossby wave activities in the westerlies influenced the troposphere above Israel after 24th August and reduced the very stable stratification. As a response, the boundary layer heights increased both over the desert and forest. We quantified the first order effect of the very large scale motions (VLSM) through a scaling law that involves the longitudinal turbulent velocity variance developed by Banerjee and Katul (2013a) and found that there are several peaks of a parameter $\alpha$ that contains the bulk effect of all kinds of VLSMs. The VLSMs are found to enhance turbulence fluxes and the nonlocal motions for both the forest and the desert. Although its main effect is to enhance the secondary circulations already existing over the desert transporting energy towards the forest.

To summarize, we have examined the existence and role of secondary circulations that exists because of large scale surface heterogeneities and possible due to some topography effects between the desert and forest by looking at proxy quantities computed from turbulence measurements. Although the campaign was conducted at a particular site, the conclusions drawn are fairly general and can be extended to other scenarios involving surface heterogeneities such as urban landscapes, agricultural fields etc. Future works will attempt to highlight a more spatially detailed picture of the turbulent structure under the interesting scenario of secondary circulations and horizontal energy transport.

*Acknowledgements.* This research was supported by the German Research Foundation (DFG) as part of the project "Climate feedbacks and benefits of semi-arid forests (CliFF)" and the project "Capturing all relevant scales of biosphere-atmosphere exchange - the enigmatic energy balance closure problem", which is funded by the Helmholtz-Association through the President's Initiative and Networking Fund, and by KIT. We suggest contacting the principal investigator Dr. Matthias Mauder (matthias.mauder@kit.edu) if one is interested in obtaining the data used in the paper.





## Appendix A:  Full form of the TKE budget

$$\frac{\partial e}{\partial t} + U\frac{\partial e}{\partial x} + V\frac{\partial e}{\partial y} + W\frac{\partial e}{\partial z} = \frac{g}{T}\left(\overline{w'T'}\right)$$
$$- \overline{u'u'}\frac{\partial U}{\partial x} - \overline{v'u'}\frac{\partial V}{\partial x} - \overline{w'u'}\frac{\partial W}{\partial x}$$
$$- \overline{u'v'}\frac{\partial U}{\partial y} - \overline{v'v'}\frac{\partial V}{\partial y} - \overline{w'v'}\frac{\partial W}{\partial y}$$
$$- \overline{u'w'}\frac{\partial U}{\partial z} - \overline{v'w'}\frac{\partial V}{\partial z} - \overline{w'w'}\frac{\partial W}{\partial z}$$

$$- \frac{\partial\left(\overline{u'e}\right)}{\partial x} - \frac{\partial\left(\overline{v'e}\right)}{\partial y} - \frac{\partial\left(\overline{w'e}\right)}{\partial z} - \frac{1}{\rho}\frac{\partial\left(\overline{u'p'}\right)}{\partial x} - \frac{1}{\rho}\frac{\partial\left(\overline{v'p'}\right)}{\partial y} - \frac{1}{\rho}\frac{\partial\left(\overline{w'p'}\right)}{\partial z} - \epsilon, \quad \text{(A1)}$$

Thus to be consistent with equation 2, all the terms in equation A1 that cannot be evaluated using one point measurements can be clubbed in the imbalance term, which can be described by

$$Imbalance = \frac{\partial e}{\partial t} + U\frac{\partial e}{\partial x} + V\frac{\partial e}{\partial y} + W\frac{\partial e}{\partial z}$$
$$+ \overline{u'u'}\frac{\partial U}{\partial x} + \overline{v'u'}\frac{\partial V}{\partial x} + \overline{w'u'}\frac{\partial W}{\partial x}$$
$$+ \overline{u'v'}\frac{\partial U}{\partial y} + \overline{v'v'}\frac{\partial V}{\partial y} + \overline{w'v'}\frac{\partial W}{\partial y}$$
$$+ \overline{v'w'}\frac{\partial V}{\partial z} + \overline{w'w'}\frac{\partial W}{\partial z}$$

$$+ \frac{\partial\left(\overline{u'e}\right)}{\partial x} + \frac{\partial\left(\overline{v'e}\right)}{\partial y} + \frac{\partial\left(\overline{w'e}\right)}{\partial z} + \frac{1}{\rho}\frac{\partial\left(\overline{u'p'}\right)}{\partial x} + \frac{1}{\rho}\frac{\partial\left(\overline{v'p'}\right)}{\partial y} + \frac{1}{\rho}\frac{\partial\left(\overline{w'p'}\right)}{\partial z}. \quad \text{(A2)}$$

Thus if no assumptions or idealizations are invoked, the imbalance of the commonly used operational TKE budget (equation 2) consists of TKE tendency, advection, shear production, TKE flux divergence and pressure velocity interactions. Using an array of sonics in each direction will enable determination of all these terms. However, as evident from the myriad of terms contributing to the imbalance, it is difficult to determine what degree of assumptions of homogeneity in which direction are sufficient so that certain terms can be ignored. Thus unless all terms in equation A2 can be determined, it is easier to stick to the most idealized form of equation 2 and treat all other terms as imbalances. Future work will try to determine the partitioning of advection, flux divergence and the other shear production terms contributing to TKE budget imbalances in presence of heterogeneities.

## Appendix B:  Further evidence of secondary circulation

Figure shows mean vertical velocity $W$ above the forest and the desert averaged over all observations using the Doppler LiDARS. Note that there is a mean updraft ($W > 0$) above the forest and a mean downdraft above the desert ($W < 0$). This suggests that there exists a mean secondary circulation and supports the findings and observations in the manuscript.





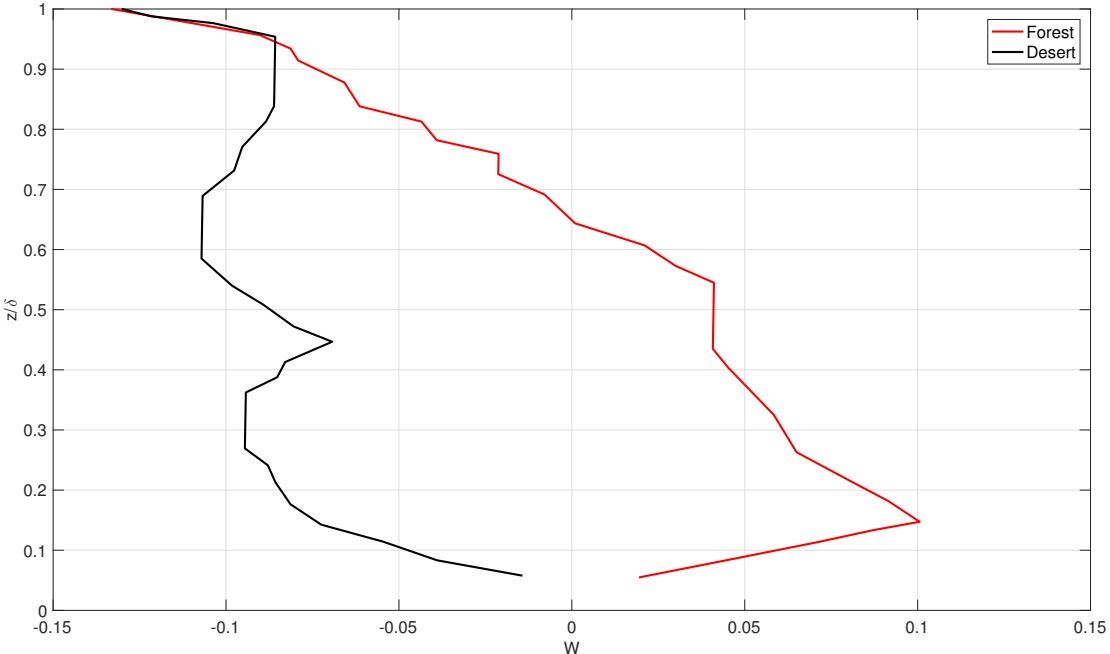

**Figure 9.** Mean vertical velocity $ms^{-1}$ above the forest and the desert as measured by LiDARs.

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
