# Peer review of "Turbulent transport of energy across a forest and a semi-arid shrubland."

_Atmospheric Chemistry and Physics, 2017_

## Referee Comment (RC1) · G. Bohrer (Referee) · 9 May 2017

P4L7 - Prognostic models predict the result in a future timestep (relative to the timestamp of observations they ingest). I think you mean here "diagnostically". P4L9 – I think you somewhat misrepresent the meaning of prognostic and diagnostic models. The difference between the two is that diagnostic model does not include a time evolution. Neither of your terms requires time evolution. Please remove the terms "prognostic" and "diagnostic". I think the best rems to use here with be "directly" and "indirectly". Also see https://earthscience.stackexchange.com/questions/924/model-types-robust-diagnostic-versus-prognostic for a good explanation.

P4 where did eq. 4 come from (it is not in Banerjee et al 2016)? And how come it does not include the roughness length?

P4L20 I recommend making this a numbered equation (the new eq 5), as this is a key component of your calculation, and you don't want to make the reader fish it out of the inline.

P4L26 Can you show the results of this regression (perhaps in an appendix)? What was its $R^2$? As you can use a whole range or r values to calculate epsilon, how did you actually do it? Picked a particular r? using the average with all possible r values (given your observation timestep and wind speed) within the 0.2-2 m range? Please add an equation stating the exact and complete formulation of epsilon the way you actually calculated it.

Figure 2 – I assume you mean the half-hourly means (or is it the hourly? Daily?) Please state it in the caption.

P5L6 (and in the description of all other figures) in "thicker" and "thinner" lines, I assume you mean "black" and "red" lines?

P7L3-6 this entire section (and similar sections that follow each of your figures) belongs in the figure caption and not in the text. You should move this to the captions of figs 2 and 3, and start section 3.1 stating: "Our observations show that the desert in associated with higher wind speed …(Fig. 2)…". I have the exact same problem with the first few line of section 3.2. Also, P10L7-11 should be removed (it is already in the caption). These are just examples, the same problem exist in many in other places.

P7L9 I totally do not agree that the increase of uâĹŮ over the desert after 24th August "can be attributed to mesoscale motions appearing over the region". I think that this is a very simple and direct result of the change in tower height. I do not accept your claim (P6L11-12) that "However, the raising of the mast should not have affected the measurement of turbulent fluxes since it was done within the constant flux layer" - Obviously, and as clearly expressed in your observations - it did.

P8L6 "however, after 24th August, the levels of w'w'…" Similarly, it is rather easy to

claim that it is due to changing the tower height. As the vertical profiles of w'w' are different between the desert and forest (due to roughness length differences), the observed differences between w'w' are a function of observation height. Apparently at 15 m above the desert and 19 m above the forest are high enough to be at the "constant flux layer", the vertical profiles of TKE (u'u' + w'w') converge. However, when you observed at lower elevation, and apparently below the constant flux layer, your data show clear differences in w'w'. As currently stated, without explicitly reminding the reader about the elevation change at that exact date, this statement is highly misleading, especially as it is immediately followed by "Thus. . ." (next sentence, L7).

Further in the same point: P9L14 "Although the effect of the large scale structure after 24th August seems to dampen the [dissipation] over the desert while its effects on the [dissipation] over the forest are not very conspicuous." Here, again, it is rather clear to me that you record less TKE dissipation when you are further from the ground and above the roughness sub-layer.

One strong argument for observed changes after Aug 24 being tower-height effects rather than change of forcing is that you only observe changes in the desert after the 24th, while the forest observation keep a rather consistent dynamics. You only changed the height of the desert tower, however, a change of forcing should be apparent over both forest and desert.

Fig 4 – what is "full TKE production"? You did not define such term, and if it is the e from eq 1, your data does not allow calculating it. I guess it is the sum of the mechanical and shear production terms. Please state it explicitly and do not call it "full TKE".

ALL figures - Please list in the caption the exact same symbols you used on the figures' y axes, so it is easier to understand what they are, and which is which. Currently you either ignore the symbols (e.g. fig 4), or provide a different version of the symbols on the caption than what is listed on the axes (e.g. fig 7 top 3 panels).

P9L7 remove "also". You already say "and"

P9L12 "huge" is a very subjective term. Perhaps "significant" (if you tested it) or "large" or simply "a" difference (can you calculate and state the % difference?)

Fig 5 – Explain what are the blue lines, and in the caption or on the figures (as in fig 6) provide the regression statistics (R^2, significance P) for the trend lines (blue?) that you are plotting.

Fig 6 Provide also the significance P.

P13L4 I do not understand why a larger integral eddy time scale over the desert is an indicator of "the transport by secondary circulations above the desert." I think it is indicative of buoyant production of turbulence, which generates larger eddies than shear production.

P14 – Please combine eq 8-10 to a single equation that relates sigma_u/u* to alpha. It is easy to see that eq 10 is totally redundant (you are re-assigning a fixed number) , and neither eq 8 or 9 are too complicated to allow direct substitution (B1 is a simple additive term in eq 8).

P14L11 How do you determine that "The data over the desert is found to be ill conditioned to compute alpha"? I think it'll be more accurate to say that this empirical formulation was originally derived for forests (using data from forest flux towers) and therefore, the values of $A\_1$ and $C"\_k$ for the desert are unknown.

Fig 8 – draw a dashed line for alpha=1 (but, as you can see below, I rather you removed this figure altogether)

Section 3.5 – I totally do not understand what you learn from the VLSM analysis (shown in bottom panel fig 8). During the entire section, you explain how to calculate alpha, and provide excuses for not calculating it over the desert, and not being unable to use it to show sea breezes and other obvious large scale circulation patterns. The only actual informative stamen you make about VLSM is that "there are a number of large peaks of $\alpha > 1$ after 24th August which confirms the presence of VLSM and

supports the interpretations of previous findings in this manuscript". I need to point out that there is presence of large peaks also before 8/24. In fact, larger (Aug 15 is the largest peak) and more (especially if you bundle up the adjacent peaks on the morning of Aug 27) peaks are present before you changed the tower height. Later, in the conclusions section (bullet point 4) you state that "The VLSMs are found to enhance turbulence fluxes and the nonlocal motions for both the forest and the desert. Although its main effect is to enhance the secondary circulations already existing over the desert transporting energy towards the forest." How do you reach this conclusion? Did you measure the correlation between alpha and turbulent fluxes? Can you prove that it enhances the mesoscale circulation already existing? This is purely speculative. If the reason for section 3.5 and conclusion point 4 is to provide justification for all the false claims about the effect of changing the tower height – than it doesn't work. It totally doesn't make a strong case to convince me that there was not effect of tower height. However, I do not understand the insistence on this entire point. Your conclusions do not rely in any way on the tower height and all the things you show about imbalance are valid before and after Aug 24, so why get yourself into this problem. Simply point out the places where the tower height may have influenced the observations, and further point out that the imbalance and other observations from which you draw conclusions about mesoscale circulations and TKE advection are showing similar patters regardless of the tower height. I will be happy if you remove this section and the 4th point of the conclusions.

---

## Referee Comment (RC2) · Anonymous Referee #2 · 23 May 2017

This manuscript utilizes a combination of high-frequency, eddy covariance measurements coupled with two Doppler wind lidars, conducted during a 12-day summer period over a desert/forest interface, with the aim of assessing the extent of the secondary circulations previously observed at this site. Additionally, the simplified TKE budget is used to explain the discrepancies between the individual budget terms over the desert and the forest. The observed discrepancies are assigned to the presence of mesoscale secondary circulations caused by the marked heterogeneity between the two opposing landuse types. The authors analyze time series and scatterplots of relevant quantities (first, second and third order statistical moments), as well as some derived quantities (integral length scales $In_{u,w}$, CBL depth $\delta$, bulk parameter $\alpha$). The authors conclude that the TKE budget terms (especially the imbalance term $Imb$) contain signatures of the aforementioned secondary circulations.

[Figure]

The manuscript provides a genuine view of the secondary circulations over a heterogeneity interface, which are currently held responsible for the surface energy balance non-closure. Hence, the study provides an important contribution to the understanding of a long-standing issue in boundary layer meteorology. The methodology implemented by the authors is well founded and the instrumental setup is sufficient for this purpose. However, the current version of the manuscript suffers from a number of critical drawbacks that the authors have not addressed, or have addressed very poorly. The manuscript requires major revisions prior to its acceptance for publication.

**Major comments**

- **page 4, line 10**: You should cite some relevant work done on TKE budgeting, in particular pertaining to how turbulent transport terms, advection terms and the pressure correlation terms may contribute individually over the desert vs. over the forest. Be aware of what may influence the imbalance terms on which your study heavily relies (especially since your $Imb$ also inherently contains the errors from the production and dissipation terms, as stated on line 11 on this page);

- **page 4, line 16**: The fact that you are conducting a field experiment over gently sloping terrain, immediately calls into place the need for more advanced rotation techniques, and the inclusion of the directional shear term $\overline{v'w'}$ into the definition of friction velocity $u_*$ (Rotach et al, 2008; Wilson, 2008);

- **page 4, line 17**: stability parameter $\zeta$ should include the displacement height $d$, so $\zeta = (z - d)/L$;

- **page 4, line 22**: Have you tried estimating $\epsilon$ using other indirect methods, for instance the inertial dissipation method?

- **page 6, line 10**: A mobile mast? Does this mean that the mast was moving around during the 12-day period? If it was not, then please omit *mobile* because it just distracts;

- **page 6, line 11**: Are you confident enough that with being just 9 meters above the canopy top you are above the roughness sublayer? There is no mention of the roughness sublayer here, and there should be one - particularly because you are applying the flux-gradient version of Monin-Obukhov similarity theory to estimate an important TKE budget term, which becomes invalid if you are within the roughness sublayer;

- **page 6, line 11**: In my opinion, here lies the biggest weakness of this manuscript. First, the raising of the mast occurred on the 23rd, and a lot of subsequent analyses describe the different behavior that suddenly began to occur from the 24th onwards, due to a passage of a *large scale mesoscale system*. Can you show that there indeed was a large scale system present, for example by showing any before/after upper-level charts? To add to this, you briefly describe the synoptic conditions in the 4th bullet point of the Conclusion (page 16, line 16) - however that information should be moved out of the Conclusion and expanded upon with supporting figures and charts much earlier in the manuscript;

- **page 6, line 12**: You are simply invoking the constant-flux layer hypothesis without citing relevant literature which actually looked at its validity. As it happens, this hypothesis is more often violated than met. Grachev et al (2005), Nadeau et al (2013) and Babić et al (2016) are some of the studies that have done this, and found the hypothesis to be true only for certain fluxes and during limited stability conditions. In particular, Babić et al (2016) have shown that the sensible heat flux is indeed constant within the daytime surface layer, however this was not true for the momentum flux. Since their study was also conducted over a shrubland, I expect similar to hold in your case (over the desert). My concern

is that the raising of the mast by 6 meters may have partially invalidated your conclusions pertaining to the evolution of the friction velocity and consequently mechanical production term after the 24th. Since you don't have at least two levels of measurements to estimate the flux divergence, I would highly recommend to cite the relevant literature and insert your view on the potential invalidation of the constant-flux layer hypothesis, especially ways in which your results may be sensitive to assuming that this hypothesis is true.

- **page 6, line 16**: You do not mention how you have pre- and post-processed the eddy covariance data. This also goes for the lidar - you list all these technical specifications (even the serial numbers!), yet you only use the lidar data to calculate the CBL depth. It should be the other way around - the eddy covariance data should be given much larger emphasis in terms of technical specs: What type of sonic anemometers were used? What rotation procedure was applied? How did you detrend the time series? How do you justify the choice of the 30-min averaging time? If all of these are the same as in Fabian Eder's AFM paper, then at least mention this.

- **page 7, line 6**: But the lower amount of friction over the desert could simply be responsible for higher wind speeds?

- **page 7, line 10**: On the contrary, this is the perfect opportunity here to discuss the synoptic conditions before and after the 24th. Please include before/after upper-level charts to clearly elucidate the structure of synoptic influence;

- **page 7, line 13**: *gentle topography* and *strong vertical updrafts* - something that is particularly sensitive to coordinate rotation. Please specify earlier what rotation technique was applied. Wilczak et al (2001) come to mind, who have shown that even a subtle misalignment of the coordinate system may lead to large errors in momentum flux estimates;

- **page 10, line 9**: Why do you report results for stable stratification all of a sudden? The goal of the manuscript is to try and gain deeper insight into the secondary circulation that causes the DAYTIME energy underclosure, not the NIGHTTIME energy overclosure. Besides, you do not talk about stable conditions hereafter all that much anyway.

- **page 11, line 7**: There is a tendency in this paper of very easily assigning the patterns in the Imbalance term to secondary circulations, with oftentimes very far-fetched statements such as this one. Please keep speculations to a minimum when you comment about terms which contain too many variables and uncertainties that you can't directly estimate (e.g. the turbulent transport term).

- **page 11, line 30**: Using *large scale* in conjunction with *mesoscale* is counterintuitive - did you mean *large scale macroscale*? If yes, then look at my point earlier above about the need to show upper-level synoptic charts.

- **page 12, line 5**: Why suddenly involve sweeps/ejections? The time scale of these coherent structures (hairpin vortices, streaky structures, ramplike convective plumes) is much smaller (20-180 s) than those of secondary circulations (several hours). Besides, secondary circulations do not *sweep* and *eject* momentum (in the Theodorsen horseshoe sense) since they are fixed to the heterogeneity interface.

- **page 13, line 4**: The integral time scale $In_u$ is typically well correlated with the CBL depth $\delta$. But here you get the opposite: even though the CBL depths over the forest and desert are roughly equal (Fig. 8), the $In_u$ scales show the opposite behavior. Furthermore, I do not see a significant bulk difference in magnitude of the integral scales before and after the 24th, i.e. based on integral scales I would not be confident saying that there was a secondary circulation after the 24th and there was not one before the 24th. I do not see the connection you made with secondary circulations appropriately justifying this discrepancy from a physical

standpoint. It looks like you are incorrectly assigning the turnover time of CBL-scale convective thermals (on the order of less than 200 s judging from Fig. 7) to turnover time of secondary circulations (which may last for several hours). If this were true, your autocorrelation function would experience a zero-crossing at much longer time lags (which obviously it does not).

- **page 13, line 8**: From Fig. 8, it is obvious that the CBL depths $\delta$ over the forest and desert are almost the same, especially after the 24th. The forest $\delta$ is only slightly larger only on the 18th and the 19th...

- **page 14, line 9**: *Ill conditioned* in what regard?

- **page 14, line 9**: Not entirely obvious to me how the forest $\alpha$ would be representative for the desert: When I look at Eq. 8, $\sigma_u$ (Fig. 3) and $\delta$ (Fig. 8) are similar between the forest and desert, but $u_*$ is very different (Fig. 2). This seems to invalidate the justification to extend the forest $\alpha$ to the desert.

- **page 14, line 11**: There are only two instances of large $\alpha$ after the 24th, while there are four instances prior to the 24th. Hence this statement is invalid.

- **Figure 8**: You don't comment on the apparent tendency for large $\alpha$ (on the 18th, 22nd, 23rd, 24th, 27th) to occur when the CBL is still growing (mostly during morning and early afternoon hours)... Any thoughts on this?

- **page 14, line 14**: This would imply that you would see a low-frequency *bump* in the vertical velocity spectra, both before and after the 24th. I would like to see a plot of the temporal evolution of the vertical velocity variance profiles from both lidars (perhaps in the form of a time-height Hovmöller diagram?). Maybe something in there would correspond well with the large $\alpha$ instances? I'm aware that Fabian Eder already did something similar in his AFM paper, however he did it only for the 25th-27th period, so after the apparent *large scale system* passage on the 24th - not before it.

**Minor comments**

- **page 1, line 16**: avoid the use of citations in abstracts.

- **page 3, Equation 1**: $\theta$, rather than $T$, is the traditionally accepted nomenclature for potential temperature(s);

- **page 3, line 25**: replace the too-colloquial *sheer* with *large*;

- **page 4, line 7**: the word *prognostically* should come earlier in this subsentence rather than at its end;

- **page 6, line 2**: You should emphasize that the Tower 1 location is different from the one analyzed in the cited Eder et al paper.

- **page 6, line 11**: What are the displacement heights at the forest and the desert sites? See above comment about proper definition of $\zeta$;

- **Figure 1**: Please include a map scale and a terrain elevation contour line. As for the north-pointing arrow, please move it to e.g. the top left corner since I barely noticed it in its current position;

- **page 6, line 24**: So the lidar at tower 2 was working during these outage periods? Why don't you then report its $\delta$ in Fig. 8?

- **Figure 2 and the following figures**: Overbars, rather than brackets, are traditionally used for denoting temporal averages. Brackets are usually used for spatial averaging. There are some inconsistencies: you use brackets around the (co)variances, while in the text you use overbars. Please correct the relevant y-labels. Additionally, specify the x-axis as time in UTC. Finally, I would recommend

putting letters to the top corner of each subplot and then accordingly modifying the text to mirror this change.

- **Figure 3**: The momentum flux should have a minus in front of it. Having it without one implies that there is a momentum source and mechanical destruction of turbulence (assuming a log law) - which is not in line with the rest of the analyses (where you do indeed have a momentum sink and a corresponding mechanical production of turbulence);

- **page 7, line 6**: The sentence *Thicker line indicates desert and thinner line indicates forest* is a remnant from a prior version of the manuscript before you replaced the thin line with a red line. Remove or modify this sentence.

- **page 9, line 5**: The sentence *Thicker line indicates desert and thinner line indicates forest* is a remnant from a prior version of the manuscript before you replaced the thin line with a red line. Remove or modify this sentence.

- **page 9, line 8**: The start of the sentence *Buoyant TKE production over the forest is slightly larger over the forest...* is unclear. Please rephrase.

- **page 10, line 2**: Replace *on the desert* with *over the desert*.

- **page 10, line 4**: Replace *indicting* with *indicating*.

- **page 10, line 4**: It would be quite instructive to calculate the Pearson correlation coefficient between the two Imbalance terms. Also adding a Imb/forest vs. Imb/desert scatterplot to Fig. 6 would be another way of expressing this;

- **Figure 5**: I cannot tell the range extent in the stability parameter in some of the subplots... Please make the x-labelticks more numerous.

- **page 11, line 6**: Be careful with wording and speculations here - sounds like you are aiming at studying the turbulent transport term (which naturally you cannot estimate in your case);

- **page 12, Eqs 6 and 7**: Is there a reason for not including the $2\epsilon_{uw}$ and $2\epsilon_{wT}$ dissipation terms here?

- **page 12, line 8**: *...opposite in nature...* sounds ambiguous. Consider rephrasing (for instance *...opposite in sign...*).

- **Figure 8**: In the spirit of Figs. 2-7+9, please replace the thin black line with a solid red line.

- **Figure 8**: Why interpolate $\delta$ on the 21st for the forest, when you don't do it anywhere else in the figure?

- **Figure 8**: Transform the y-axis into a logarithmic one, given the prevalence of small $\alpha$.

- **Figure 9**: Please consider scaling the averaged vertical velocity on the x-axis with the average Deardorff convective velocity scale $w_*$.

References
1. Babić, Nevio, Željko Večenaj, and Stephan FJ De Wekker. "Flux–Variance Similarity in Complex Terrain and Its Sensitivity to Different Methods of Treating Non-stationarity." Boundary-layer meteorology 159.1 (2016): 123-145.
2. Grachev, Andrey A., et al. "Stable boundary-layer scaling regimes: the SHEBA data." Boundary-Layer Meteorology 116.2 (2005): 201-235.
3. Nadeau, Daniel F., et al. "Similarity scaling over a steep alpine slope." Boundary-layer meteorology 147.3 (2013): 401-419.
4. Rotach, Mathias W., et al. "Boundary layer characteristics and turbulent exchange

mechanisms in highly complex terrain." Acta Geophysica 56.1 (2008): 194-219.

5. Wilczak, James M., Steven P. Oncley, and Steven A. Stage. "Sonic anemometer tilt correction algorithms." Boundary-Layer Meteorology 99.1 (2001): 127-150.

6. Wilson, J. D. "Monin-Obukhov functions for standard deviations of velocity." Boundary-layer meteorology 129.3 (2008): 353-369.

---

## Author Comment (AC1) · 1 Feb 2018

**Response to reviewer 1**

We thank Prof. Bohrer for the constructive comments and suggestions.

**P4L7 - Prognostic models predict the result in a future timestep (relative to the times- tamp of observations they ingest). I think you mean here "diagnostically".**

Corrected.

**P4L9 – I think you somewhat misrepresent the meaning of prognostic and diagnostic models. The difference between the two is that diagnostic model does not include a time evolution. Neither of your terms requires time evolution. Please remove the terms "prognostic" and "diagnostic". I think the best rems to use here with be "directly" and "indirectly". Also see https://earthscience.stackexchange.com/questions/924/model-types- robust-diagnostic-versus-prognostic for a good explanation.**

Thanks for pointing this out. We have removed the terms prognostic and diagnostic. The section now reads like this: "*Under these constraints, a strategy is needed to evaluate the TKE budget. The dominant mechanical production term, the buoyant production/destruction term and the dissipation term will be evaluated directly from the data. The residual of the TKE budget will be described as the imbalance as per equation 3 which would contain the effects of advection and transport terms.*"

**P4 where did eq. 4 come from (it is not in Banerjee et al 2016)? And how come it does not include the roughness length?**

The equation is defined inline in Banerjee et al., 2016 after equation 3. However, two new references are added, where they are defined more explicitly (Li et al., 2016 and Kaimal and Finnigan, 1994). The roughness length comes in the equation for the profile of the mean longitudinal velocity, which can be derived by integrating equation 4. The roughness length comes as the lower integration constant. The gradient of velocity should be independent of the surface boundary condition.

**P4L20 I recommend making this a numbered equation (the new eq 5), as this is a key component of your calculation, and you don't want to make the reader fish it out of the inline.**

Agreed and changed to numbered equation.

**P4L26 Can you show the results of this regression (perhaps in an appendix)? What was its R^2? As you can use a whole range or r values to calculate epsilon, how did you actually do**

**it? Picked a particular r? using the average with all possible r values (given your observation timestep and wind speed) within the 0.2-2 m range? Please add an equation stating the exact and complete formulation of epsilon the way you actually calculated it.**

Since it has been a standard technique, it is not repeated in the main text and just the references are added. It is only discussed in the letter following Salesky 2013.. As mentioned in the text, the scaling relation used is $D_{uu}(r) = C_u \epsilon^{2/3} r^{2/3}$,

where $c_2 \approx 4.017 c_k$, $c_k = 18 c_e/55$, and $c_e = 1.5$ is the Kolmogorov constant for the inertial range of the TKE spectrum E(k). Our estimate of $\epsilon$ was calculated by a linear regression to the compensated second-order structure function $r_1^{-2/3} D_{11}(r_1)$, i.e., $r_1^{-2/3} D_{11}(r_1) = c_2 \epsilon^{2/3} = a r_1 + b$,

[Figure]

using values of r1 in the range $0.2 \leq r1 \leq 2.0$ m. The lower limit imposed on r1 ensures that distortions from the sonic anemometer finite path length are negligible. The upper limit on r1 is selected so as to ensure at least one decade of scales is available in the determination of ε. The regression coefficient b was used to obtain an estimate of the dissipation rate (i.e., $\epsilon = (b/c_2)^{3/2}$); the coefficient a must be nearly zero if the data follow inertial-range scaling. The top panel of the attached figure shows a sample half hour high frequency time series. The middle panel shows the 2/3 scaling fit to the structure function and the third panel shows the extracted dataset between the r range 0.2-2.

**Figure 2 – I assume you mean the half-hourly means (or is it the hourly? Daily?) Please state it in the caption.**

Half hourly, mentioned in caption now.

**P7L6 (and in the description of all other figures) in "thicker" and "thinner" lines, I assume you mean "black" and "red" lines?**

Yes, corrected. Thanks for pointing this out. We also corrected the same mistake on P9L5. These two instances remained after we changed the thick - thin scheme which was used in the earlier version of the manuscript.

**P7L3-6 this entire section (and similar sections that follow each of your figures) be- longs in the figure caption and not in the text. You should move this to the captions of figs 2 and 3, and start section 3.1 stating: "Our observations show that the desert in associated with higher wind speed . . .(Fig. 2). . .". I have the exact same problem with the first few line of section 3.2. Also, P10L7-11 should be removed (it is already in the caption). These are just examples, the same problem exist in many in other places.**

We have faced instances where reviewers had felt uncomfortable without figure descriptions in the main text body although they were in the caption - since this is a subjective editorial issue, we are not changing this style at this time.

**P7L9 I totally do not agree that the increase of $u_*$ over the desert after 24th August "can be attributed to mesoscale motions appearing over the region". I think that this is a very simple and direct result of the change in tower height. I do not accept your claim (P6L11-12) that "However, the raising of the mast should not have affected the measurement of turbulent fluxes since it was done within the constant flux layer" - Obviously, and as clearly expressed in your observations - it did.**

Accepted. Changed to *"This can be attributed to the raising of the tower height"*. Also deleted the sentence : *"However, the raising of the mast should not have affected the measurement of turbulent fluxes since it was done within the constant flux layer"*.

**P8L6 "however, after 24th August, the levels of w'w'. . ." Similarly, it is rather easy to claim that it is due to changing the tower height. As the vertical profiles of w'w' are different between the desert and forest (due to roughness length differences), the observed differences between w'w' are a function of observation height. Apparently at 15 m above the desert and 19 m above the forest are high enough to be at the "constant flux layer", the vertical profiles of TKE (u'u' + w'w') converge. However, when you observed at lower elevation, and apparently below the constant flux layer, your data show clear differences in w'w'. As currently stated, without explicitly reminding the reader about the elevation change at that exact date, this statement is highly misleading, especially as it is immediately followed by "Thus. . ." (next sentence, L7).**

Accepted. The sentences describing the effect of large scale structures for $\overline{u'u'}$ *and* $\overline{u'w'}$ *are removed as well*. The section is replaced by:

*The vertical velocity variance $\overline{w'w'}$ over the forest is higher than its desert counterpart, however, after 24th August, the levels of $\overline{w'w'}$ over desert increases as well and become similar to the forest. It is due to changing the tower height. As the vertical profiles of $\overline{w'w'}$ are different between the desert and forest (due to roughness length differences), the observed differences*

*between $\overline{w'w'}$ are a function of observation height. At 15 m above the desert and 19 m above the forest are high enough to be at the "constant flux layer", the vertical profiles of TKE ($\overline{u'u'}+\overline{w'w'}$) converge. However, when observed at a lower elevation, and below the constant flux layer, the data show clear differences in $\overline{w'w'}$.*

**Further in the same point: P9L14 "Although the effect of the large scale structure after 24th August seems to dampen the [dissipation] over the desert while its effects on the [dissipation] over the forest are not very conspicuous." Here, again, it is rather clear to me that you record less TKE dissipation when you are further from the ground and above the roughness sub-layer. One strong argument for observed changes after Aug 24 being tower-height effects rather than change of forcing is that you only observe changes in the desert after the 24th, while the forest observation keep a rather consistent dynamics. You only changed the height of the desert tower, however, a change of forcing should be apparent over both forest and desert.**

Agreed. This section is rewritten as : "*A smaller TKE dissipation is recorded when the measurement location is further from the ground and above the roughness sub-layer. One strong argument for observed changes after Aug 24 being tower-height effects rather than change of any large scale forcing is that changes in the desert are observed only after the 24th, while the forest observations maintain a rather consistent dynamics.*"

**Fig 4 – what is "full TKE production"? You did not define such term, and if it is the e from eq 1, your data does not allow calculating it. I guess it is the sum of the mechanical and shear production terms. Please state it explicitly and do not call it "full TKE".**

It is defined as the summation of mechanical and buoyant TKE production.

**ALL figures - Please list in the caption the exact same symbols you used on the figures' y axes, so it is easier to understand what they are, and which is which. Currently you either ignore the symbols (e.g. fig 4), or provide a different version of the symbols on the caption than what is listed on the axes (e.g. fig 7 top 3 panels).**

Please note that all terms are listed on the section of the text describing the figure. This way, the caption and the figure description are not exactly the same, referring to Your earlier point.

**P9L7 remove "also". You already say "and"**

Removed.

**P9L12 "huge" is a very subjective term. Perhaps "significant" (if you tested it) or "large" or simply "a" difference (can you calculate and state the % difference?)**

Agreed and removed huge. The % difference changes with time, following the exact same trend as the actual terms. replaced by: "*It also indicates that mechanical forcing, and not buoyancy makes a difference (mechanical production is higher by approximately an order of magnitude than buoyant production) in the turbulence generation over the desert and the forest*".

**Fig 5 – Explain what are the blue lines, and in the caption or on the figures (as in fig 6) provide the regression statistics (R^2, significance P) for the trend lines (blue?) that you are plotting.**

This figure is now removed as we realized that it is not conveying much more information other than what is already there in figure 6.

**Fig 6 Provide also the significance P.**

p : 0.05.

**P13L4 I do not understand why a larger integral eddy time scale over the desert is an indicator of "the transport by secondary circulations above the desert." I think it is indicative of buoyant production of turbulence, which generates larger eddies than shear production.**

Agreed and corrected.

**P14 – Please combine eq 8-10 to a single equation that relates sigma_u/u* to alpha. It is easy to see that eq 10 is totally redundant (you are re-assigning a fixed number) , and neither eq 8 or 9 are too complicated to allow direct substitution (B1 is a simple additive term in eq 8).**

This section is now removed. We agree with Your argument.

**P14L11 How do you determine that "The data over the desert is found to be ill conditioned to compute alpha"? I think it'll be more accurate to say that this empirical formulation was originally derived for forests (using data from forest flux towers) and therefore, the values of A_1 and C"_k for the desert are unknown.**

This section is now removed.

**Fig 8 – draw a dashed line for alpha=1 (but, as you can see below, I rather you removed this figure altogether) Section 3.5 – I totally do not understand what you learn from the VLSM analysis (shown in bottom panel fig 8). During the entire section, you explain how to calculate alpha, and provide excuses for not calculating it over the desert, and not being unable to use it to show sea breezes and other obvious large scale circulation patterns. The only actual informative stamen you make about VLSM is that "there are a number of large peaks of ⌡ > 1 after 24th August which confirms the presence of VLSM and supports the interpretations of previous findings in this manuscript". I need to point out that there is presence of large peaks also before 8/24. In fact, larger (Aug 15 is the largest peak) and more (especially if you bundle up the adjacent peaks on the morning of Aug 27) peaks are present before you changed the tower height. Later, in the conclusions section (bullet point 4) you state that "The VLSMs are found to enhance turbulence fluxes and the nonlocal motions for both the forest and the desert. Although its main effect is to enhance the secondary circulations already existing over the desert transporting energy towards the forest." How do you reach this conclusion? Did you measure the correlation between alpha and turbulent fluxes? Can you prove that it enhances the mesoscale circulation already existing? This is purely speculative. If the reason for section 3.5 and conclusion point 4 is to provide justification for all the false claims about the effect of changing the tower height – than it doesn't work. It totally doesn't make a strong case to convince me that there was not effect of tower height. However, I do not understand the insistence on this entire point. Your conclusions do not rely in any way on the tower height and all the things you show about imbalance are valid before and after Aug 24, so why get yourself into this problem. Simply point out the places where the tower height may have influenced the observations, and further point out that the imbalance and other observations from which you draw conclusions about mesoscale circulations and TKE advection are showing similar patters regardless of the tower height. I will be happy if you remove this section and the 4th point of the conclusions.**

Agreed. We have now removed the section and the 4th point of the calculation. Earlier discussions have now also pointed out the changes after 24th occurs due to tower height change.

---

## Author Comment (AC2) · 1 Feb 2018

**This manuscript utilizes a combination of high-frequency, eddy covariance measurements coupled with two Doppler wind lidars, conducted during a 12-day summer period over a desert/forest interface, with the aim of assessing the extent of the secondary circulations previously observed at this site. Additionally, the simplified TKE budget is used to explain the discrepancies between the individual budget terms over the desert and the forest. The observed discrepancies are assigned to the presence of mesoscale secondary circulations caused by the marked heterogeneity between the two opposing land use types. The authors analyze time series and scatterplots of relevant quantities (first, second and third order statistical moments), as well as some derived quantities (integral length scales $In_{u,w}$, CBL depth $\delta$, bulk parameter $\alpha$). The authors conclude that the TKE budget terms (especially the imbalance term Imb) contain signatures of the aforementioned secondary circulations.**

**The manuscript provides a genuine view of the secondary circulations over a heterogeneity interface, which are currently held responsible for the surface energy balance non-closure. Hence, the study provides an important contribution to the understanding of a long-standing issue in boundary layer meteorology. The methodology implemented by the authors is well founded and the instrumental setup is sufficient for this purpose. However, the current version of the manuscript suffers from a number of critical drawbacks that the authors have not addressed, or have addressed very poorly. The manuscript requires major revisions prior to its acceptance for publication.**

We thank the reviewer for the constructive comments and suggestions.

**Major comments**

**page 4, line 10: You should cite some relevant work done on TKE budgeting, in particular pertaining to how turbulent transport terms, advection terms and the pressure correlation terms may contribute individually over the desert vs. over the forest. Be aware of what may influence the imbalance terms on which your study heavily relies (especially since your Imb also inherently contains the errors from the production and dissipation terms, as stated on line 11 on this page)**

Not many instances were found in the literature where the nature of turbulent transport were studied across large scale surface roughness heterogeneities, except for Nadeau 2011 and Yue 2015. These references are now added.

**page 4, line 16: The fact that you are conducting a field experiment over gently sloping terrain, immediately calls into place the need for more advanced rotation techniques, and**

the inclusion of the directional shear term $\overline{v'w'}$ into the definition of friction velocity $u_*$ (Rotach et al, 2008; Wilson, 2008);

$u*$ now contains $\overline{v'w'}$ in its calculation.

**page 4, line 17: stability parameter $\zeta$ should include the displacement height d, so $\zeta = (z-d)/L$;**

It was already calculated using the displacement length d, the text is now corrected.

**page 4, line 22: Have you tried estimating $\epsilon$ using other indirect methods, for instance the inertial dissipation method?**

No, the structure function is used as it usually shows a more robust scaling relation compared to the spectral (inertial dissipation) method since it is calculated in the real space. Moreover, the scaling relation (2/3) in the structure function method can be translated into the scaling relation (-5/3) in spectral space - so ultimately there is not much difference between the two.

**page 6, line 10: A mobile mast? Does this mean that the mast was moving around during the 12-day period? If it was not, then please omit mobile because it just distracts;**

Removed.

**page 6, line 11: Are you confident enough that with being just 9 meters above the canopy top you are above the roughness sublayer? There is no mention of the roughness sublayer here, and there should be one - particularly because you are applying the flux-gradient version of Monin-Obukhov similarity theory to estimate an important TKE budget term, which becomes invalid if you are within the roughness sublayer;**

[Figure]

This is a great point and it was already discussed in the first round of revision. As observed from the figure taken from of the roughness sublayer correction function from the paper (figure 2a): Harman, I. N., and J. J. Finnigan, 2007, A simple unified theory for flow in the canopy and roughness sublayer. Boundary-Layer Meteorol., 123, 339–363, its value is about 1 at z/h=2

which is the case in this campaign. This correction function $\phi_{mc}$ is multiplicative to the original stability correction function $\phi_m$. So its value being 1, this is not included. Moreover, it also justifies the fact that we are above the roughness sub layer for both heights 9 m and 18m. We agree that this was not articulated well in the text before. Now it is mentioned.

**page 6, line 11: In my opinion, here lies the biggest weakness of this manuscript. First, the raising of the mast occurred on the 23rd, and a lot of subsequent analyses describe the different behavior that suddenly began to occur from the 24th onwards, due to a passage of a large scale mesoscale system. Can you show that there indeed was a large scale system present, for example by showing any before/after upper-level charts? To add to this, you briefly describe the synoptic conditions in the 4th bullet point of the Conclusion (page 16, line 16) - however that information should be moved out of the Conclusion and expanded upon with supporting figures and charts much earlier in the manuscript;**

Following the suggestion of reviewer 1, we have removed this sentence. We realized that changing the mast height was indeed responsible for the changes observed after 24th. We have also removed section 3.5 and all discussions of the passage of the large scale structures from the conclusion as well. Changes in individual statistics have been explained in conjunction with the raising of the mast as reviewer 1 suggested. Please see response to reviewer 1.

**page 6, line 12: You are simply invoking the constant-flux layer hypothesis with- out citing relevant literature which actually looked at its validity. As it happens, this hypothesis is more often violated than met. Grachev et al (2005), Nadeau et al (2013) and Babic´ et al (2016) are some of the studies that have done this, and found the hypothesis to be true only for certain fluxes and during limited stability conditions. In particular, Babic´ et al (2016) have shown that the sensible heat flux is indeed constant within the daytime surface layer, however this was not true for the momentum flux. Since their study was also conducted over a shrubland, I expect similar to hold in your case (over the desert). My concern is that the raising of the mast by 6 meters may have partially invalidated your conclusions pertaining to the evolution of the friction velocity and consequently mechanical production term after the 24th. Since you don't have at least two lev- els of measurements to estimate the flux divergence, I would highly recommend to cite the relevant literature and insert your view on the potential invalidation of the constant-flux layer hypothesis, especially ways in which your results may be sensitive to assuming that this hypothesis is true.**

It is a valid suggestion. To avoid the confusion, we have removed the sentence altogether. As pointed out by reviewer 1, certain changes can indeed be attributed to the change in mast height. As we have noted later: "*The vertical velocity variance $\overline{w'w'}$ over the forest is higher than its desert counterpart, however, after 24th August, the levels of $\overline{w'w'}$ over desert increases as well*

*and become similar to the forest. It is due to changing the tower height. As the vertical profiles of $\overline{w'w'}$ are different between the desert and forest (due to roughness length differences), the observed differences between $\overline{w'w'}$ are a function of observation height. At 15 m above the desert and 19 m above the forest are high enough to be at the "constant flux layer", the vertical profiles of TKE ($\overline{u'u'}+\overline{w'w'}$) converge. However, when observed at a lower elevation, and below the constant flux layer, the data show clear differences in $\overline{w'w'}$.*

*also,*

*"A smaller TKE dissipation is recorded when the measurement location is further from the ground and above the roughness sub-layer. One strong argument for observed changes after Aug 24 being tower-height effects rather than change of any large scale forcing is that changes in the desert are observed only after the 24th, while the forest observations maintain a rather consistent dynamics."*

**page 6, line 16: You do not mention how you have pre- and post-processed the eddy covariance data. This also goes for the lidar - you list all these technical specifications (even the serial numbers!), yet you only use the lidar data to calculate the CBL depth. It should be the other way around - the eddy covariance data should be given much larger emphasis in terms of technical specs: What type of sonic anemometers were used? What rotation procedure was applied? How did you detrend the time series? How do you justify the choice of the 30-min averaging time? If all of these are the same as in Fabian Eder's AFM paper, then at least mention this.**

We have simply mentioned that the details of the EC method are similar to Eder 2015 paper. More details are added on the rotation technique.

**page 7, line 6: But the lower amount of friction over the desert could simply be responsible for higher wind speeds?**

Noted and added.

**page 7, line 10: On the contrary, this is the perfect opportunity here to discuss the synoptic conditions before and after the 24th. Please include before/after upper-level charts to clearly elucidate the structure of synoptic influence;**

Following the argument from reviewer 1, This is changed to : *"This can be attributed to the raising of the tower height"*.

**page 7, line 13: gentle topography and strong vertical updrafts - something that is particularly sensitive to coordinate rotation. Please specify earlier what rotation technique was applied. Wilczak et al (2001) come to mind, who have shown that even a subtle misalignment of the coordinate system may lead to large errors in momentum flux estimates;**

Discussion on rotation technique added.

**page 10, line 9: Why do you report results for stable stratification all of a sudden? The goal of the manuscript is to try and gain deeper insight into the secondary circulation that causes the DAYTIME energy underclosure, not the NIGHTTIME energy overclosure. Besides, you do not talk about stable conditions hereafter all that much anyway.**

We realize that figure 5 is not conveying more information, so it is removed altogether. The same information can be extracted from figure 6.

**page 11, line 7: There is a tendency in this paper of very easily assigning the patterns in the Imbalance term to secondary circulations, with oftentimes very far-fetched statements such as this one. Please keep speculations to a minimum when you comment about terms which contain too many variables and uncertain- ties that you can't directly estimate (e.g. the turbulent transport term).**

This is a great point and was also raised by reviewer 1 in the previous review. To clarify this issue, we resort to principal component analysis (PCA) which shows the relationships between different variables in a multidimensional data space (not shown in the paper). The first panel shows the centered and scaled data. The second panel shows the total amount of data variability explained by the principal components. As observed, top 2 principal components explain about 80% of the data variability. The 3rd panel shows the 2d biplot and the 4th panel shows the 3d biplot. The angles between the vectors indicate the degree of correlation between the variables. An angle of zero degrees or 180 degrees indicates perfect correlation and orthogonality between the vectors indicate zero correlation.

The biplot confirms that the desert production is highly correlated with the forest production as they should have the same forcing. The desert production is also correlated with the TKE imbalance over the forest. However, the desert production has weaker correlation with desert imbalance. Moreover, the desert imbalance has high correlation with the forest production. The imbalance over desert and forest have almost zero correlation. This indicates that the large scale mostly thermal structures are transported from the desert to the forest. Not all the transport over the desert is generated by the desert production and the large scale nonlocal structures contribute

to the transport over the desert, which creates additional forcing over the forest. Thus the high correlation between the desert imbalance and forest production is retained. However, the additional correlations between the desert production and desert imbalance, as well as desert production and forest production are added and acknowledged in the text.

**page 11, line 30: Using large scale in conjunction with mesoscale is counterintuitive - did you mean large scale macroscale? If yes, then look at my point earlier above about the need to show upper-level synoptic charts.**

As discussed earlier, we have removed all discussions on mesoscale motions.

**page 12, line 5: Why suddenly involve sweeps/ejections? The time scale of these coherent structures (hairpin vortices, streaky structures, ramplike convective plumes) is much smaller (20-180 s) than those of secondary circulations (several hours). Besides, secondary circulations do not sweep and eject momentum (in the Theodorsen horseshoe sense) since they are fixed to the heterogeneity interface.**

Triple moments have been shown to be connected with sweep-ejection motions (Nakagawa and Nezu, 1977; Raupach et al., 1986; Cava et al., 2006; Katul et al., 2013; Banerjee et al., 2017a). We don't mean to say that the secondary circulations are causing these motions, but the net transport of turbulent energy is causing them.

**page 13, line 4: The integral time scale In$_u$ is typically well correlated with the CBL depth . But here you get the opposite: even though the CBL depths over the forest and desert are roughly equal (Fig. 8), the In$_u$ scales show the opposite behavior. Furthermore, I do not see a significant bulk difference in magnitude of the integral scales before and after the 24th, i.e. based on integral scales I would not be confident saying that there was a secondary circulation after the 24th and there was not one before the 24th. I do not see the connection you made with secondary circulations appropriately justifying this discrepancy from a physical standpoint. It looks like you are incorrectly assigning the turnover time of CBL-scale convective thermals (on the order of less than 200 s judging from Fig. 7) to turnover time of secondary circulations (which may last for several hours). If this were true, your autocorrelation function would experience a zero-crossing at much longer time lags (which obviously it does not).**

Agreed. This sentence is removed and replaced by what was suggested by reviewer 1: "*More interesting is the observation that the integral time scales for the eddies above the desert are larger than the forest- both of which increase after 24th. This is another indicator of buoyant production of turbulence, which generates larger eddies than shear production*".

**page 13, line 8: From Fig. 8, it is obvious that the CBL depths over the forest and desert are almost the same, especially after the 24th. The forest is only slightly larger only on the 18th and the 19th...**

This section is now removed.

**page 14, line 9: Ill conditioned in what regard?**

This section is now removed.

**page 14, line 9: Not entirely obvious to me how the forest $\alpha$ would be representative for the desert: When I look at Eq. 8, $\sigma_u$ (Fig. 3) and $\delta$ (Fig. 8) are similar between the forest and desert, but $u_*$ is very different (Fig. 2). This seems to invalidate the justification to extend the forest $\alpha$ to the desert.**

This section is now removed.

**page 14, line 11: There are only two instances of large $\alpha$ after the 24th, while there are four instances prior to the 24th. Hence this statement is invalid.**

This section is now removed.

**Figure 8: You don't comment on the apparent tendency for large $\alpha$ (on the 18th, 22nd, 23rd, 24th, 27th) to occur when the CBL is still growing (mostly during morning and early afternoon hours)... Any thoughts on this?**

This figure is now removed.

**page 14, line 14: This would imply that you would see a low-frequency bump in the vertical velocity spectra, both before and after the 24th. I would like to see a plot of the temporal evolution of the vertical velocity variance profiles from both lidars (perhaps in the form of a time-height Hovmöller diagram?). Maybe something in there would correspond well with the large $\alpha$ instances? I'm aware that Fabian Eder already did something similar in his AFM paper, however he did it only for the 25th-27th period, so after the apparent large scale system passage on the 24th - not before it.**

This section is now removed.

**Minor comments**

**page 1, line 16: avoid the use of citations in abstracts.**

Removed.

**page 3, Equation 1: $\theta$, rather than T , is the traditionally accepted nomenclature for potential temperature(s);**

Since we have used T consistently in the paper and other previous papers, it is retained.

**page 3, line 25: replace the too-colloquial sheer with large;**

Replaced.

**page 4, line 7: the word prognostically should come earlier in this subsentence rather than at its end;**

As pointed out by reviewer 1, we have removed the terms prognostically and diagnostically.

**page 6, line 2: You should emphasize that the Tower 1 location is different from the one analyzed in the cited Eder et al paper.**

Mentioned.

**page 6, line 11: What are the displacement heights at the forest and the desert sites? See above comment about proper definition of $\zeta$**

There is no displacement length considered for the desert. For the forest, it is taken as 2/3rd canopy height.

**Figure 1: Please include a map scale and a terrain elevation contour line. As for the north-pointing arrow, please move it to e.g. the top left corner since I barely noticed it in its current position;**

Done.

**page 6, line 24: So the lidar at tower 2 was working during these outage periods? Why don't you then report its in Fig. 8?**

Figure 8 is now removed.

**Figure 2 and the following figures: Overbars, rather than brackets, are traditionally used for denoting temporal averages. Brackets are usually used for spatial averaging. There are some inconsistencies: you use brackets around the (co)variances, while in the text you use overbars. Please correct the relevant y- labels. Additionally, specify the x-axis as time in UTC. Finally, I would recommend putting letters to the top corner of each subplot and then accordingly modifying the text to mirror this change.**

MATLAB has been used to generate this figures, and a glitch does not allow having over bars in the labels. It has been mentioned that they convey the same thing. time in UTC is mentioned. Letters are not used since there is only one column and each has a specific ylabel.

**Figure 3: The momentum flux should have a minus in front of it. Having it with- out one implies that there is a momentum source and mechanical destruction of turbulence (assuming a log law) - which is not in line with the rest of the analyses (where you do indeed have a momentum sink and a corresponding mechanical production of turbulence);**

Corrected.

**page 7, line 6: The sentence Thicker line indicates desert and thinner line in- dicates forest is a remnant from a prior version of the manuscript before you replaced the thin line with a red line. Remove or modify this sentence.**

Corrected.

**page 9, line 5: The sentence Thicker line indicates desert and thinner line in- dicates forest is a remnant from a prior version of the manuscript before you replaced the thin line with a red line. Remove or modify this sentence.**

Corrected.

**page 9, line 8: The start of the sentence Buoyant TKE production over the forest is slightly larger over the forest... is unclear. Please rephrase.**

Corrected.

**page 10, line 2: Replace on the desert with over the desert.**

Done.

**page 10, line 4: Replace indicting with indicating.**

Corrected.

**page 10, line 4: It would be quite instructive to calculate the Pearson correla- tion coefficient between the two Imbalance terms. Also adding a Imb/forest vs. Imb/desert scatterplot to Fig. 6 would be another way of expressing this;**

The physical significance of that correlation is not well understood, so we are not adding this. Moreover, from the pca analysis shown before, these two imbalances have almost zero correlation (orthogonal to each other).

**Figure 5: I cannot tell the range extent in the stability parameter in some of the subplots... Please make the x-label ticks more numerous.**

This figure is now removed.

**page 11, line 6: Be careful with wording and speculations here - sounds like you are aiming at studying the turbulent transport term (which naturally you cannot estimate in your case);**

Rephrased to : figure 6 is used to better understand the nature of turbulent transport between the desert and the forest.

**page 12, Eqs 6 and 7: Is there a reason for not including the $\epsilon_{uw}$ and $\epsilon_{wT}$ dissipation terms here?**

We just wanted to be consistent with the other references provided.

**page 12, line 8: ...opposite in nature... sounds ambiguous. Consider rephrasing (for instance ...opposite in sign...).**

Corrected.

**Figure 8: In the spirit of Figs. 2-7+9, please replace the thin black line with a solid red line.**

Figure now removed.

**Figure 8: Why interpolate $\delta$ on the 21st for the forest, when you don't do it any- where else in the figure?**

Figure now removed.

**Figure 8: Transform the y-axis into a logarithmic one, given the prevalence of small $\alpha$.**

Figure now removed.

**Figure 9: Please consider scaling the averaged vertical velocity on the x-axis with the average Deardorff convective velocity scale $w_*$;**

We wanted to show the strength of the recirculation, so the unit is retained.

---

## Referee Report (RR1)

**Turbulent transport of energy across a forest and a semi-arid shrubland**

**Banerjee et al**

**round 1 verdict: MAJOR REVISION**

In most cases, the responses of the authors have been acceptable. Specifically with the raising of the mast over the desert on the 23rd, most of the comments and analyses have been modified to reflect this more objectively and correctly. Some of the other weaknesses have also been appropriately addressed, e.g. justifying the constant flux layer hypothesis. The authors also interpret results more cautiously when attributing the observed differences in the imbalance term between the forest and the desert, to a secondary circulation.

However, the manuscript still feels incomplete, particularly because of the heavily underutilized Doppler wind lidars. As it currently stands, these lidars are only mentioned in Appendix B - which is not even referenced in the main text at all. As such, Figure 7 contradicts the time series of $\langle W \rangle$ from the sonics - but this is not explained or even mentioned at all in the present version of the manuscript.

I propose another major revision of the manuscript, by incorporating more analyses involving the two Doppler wind lidars. In doing so, the authors have a unique opportunity to provide another view of the secondary circulations which hasn't been explored so far. I believe that such analyses will substantially increase the value of this study.

**Major comments**

- **page 4, line 20**: Although you apply planar fit now, you still don't incorporate the directional shear $\overline{v'w'}$, in the definition of the friction velocity. It's likely it won't alter the current $u_*$ values that much, but it should still be included for completeness;

- **page 5, line 28**: how was $\delta$ obtained?

- **page 7, line 1**: Is the mean vertical velocity that is plotted in Fig. 2, the vertical velocity that is rotated with the planar fit procedure?

- **Fig. 2 and Fig. 7**: This is where the major drawback of this study lies: In addition to these two figures contradicting each other, it remains unclear why that would be so since the authors do not address it at all.

  First, the authors do not compare Fig. 7 with Fig. 4a,b in Eder et al (2015, AFM).

Their LES results and your lidar results are in agreement (at least in terms of the sign of average vertical velocity).

Second, it's obvious from Fig. 7 that at the lowest range gates of both lidars, the vertical velocities converge to zero, perhaps maybe even changing signs very near the surface - which would then be in accordance with Fig. 2. However, this area is under the influence of the dead zone from the lidars, so only speculation is allowed. Even so, it remains clear from Fig. 7 that, throughout the CBL, it is the desert that experiences a downdraft and the forest is the one associated with an updraft. This is in contradiction with the general conclusion of the study, i.e. with the orientation of such a secondary circulation that the authors suggest exists (based on $Imb$ and triple-order moments analyses).

This raises another question: does the secondary circulation extend all the way to the top of the CBL or just to a certain $z/\delta$? The discrepancy explained in the previous paragraph would suggest the latter. As pointed out by the authors, an answer may lie in the vertical velocity skewness $Sk_w$. A simple analysis to do would be to, for available times that the lidars were working, obtain vertical profiles of $Sk_w$ and inspect their temporal evolution over both the desert and the forest. E.g. plotting a time-height Howmoller diagram for the whole 12 days of the campaign comes to mind. Since the forest experiences negative $Sk_w$, it would be interesting to see how far up this persists. Of course, because a third-order moment is involved, a longer averaging period is required - for instance 1 (maybe even 2) hour (thus skipping the single VAD scan that was performed in between vertical stares).

Another useful avenue that would go along the 'eddy size distribution' line-of-thinking that the authors mention in the beginning of the Conclusion, would be to spectrally decompose the lidar-derived $Sk_w$. This has been done on several occasions (Hogan et al, 2009, their Fig. 7). By doing this for each range gate, a height-frequency Howmoller diagram could be designed (for a single representative case study, not all 12 days). Such a diagram would then elucidate how each scale of motion contributes to overall $Sk_w$. I'm certain some interesting differences between the forest and the desert would become apparent.

- **page 11, line 6**: I'm still not comfortable with bringing ejections and sweeps into the mix here (as I've pointed out in my first response). I agree that they are contributors to third-order moments, but the way you describe it currently makes it seem like ejections and sweeps are related to secondary circulations - which is not true. A word of caution would help here, disclosing the fact that ejections/sweeps are a much finer-scale, smaller-scale motions than secondary circulations.

In line with the spectral decomposition of lidar-derived $Sk_w$ I mentioned earlier, I wonder if decomposing the mixed third-order moments would reveal some aspects at the low-frequency part of those cospectra? Significant deviations should be expected there,

considering the different ranges of signs (between desert and forest) that you obtained in Fig. 6.

Also, the fact that higher-order moments require longer averaging times than means and variances, makes me question how reliable the different signs between the forests and desert actually are. Sensitivity analyses of $\langle w'w'u' \rangle$, $\langle w'w'w' \rangle$ and $\langle w'w'T' \rangle$ as a function of increasing the averaging time with increments of 30 minutes, would make the results more bulletproof.

**Minor comments**

- **page 2, line 5**: having both *or* and *etc.* is redundant, keep just one of them;

- **page 4, line 16**: it's usually referenced as Monin-Obukhov Similarity Theory (instead of Stability);

- **page 8, line 7**: 23rd instead of 23th;

- **page 8, line 12**: 23rd instead of 23th;

- **page 9, line 1**: *in* the constant flux layer;

- **page 9, line 12**: replace *summation* with *sum*

- **page 10, line 3**: 23rd instead of 23th;

- **page 10, line 12**: 23rd instead of 23th;

- **page 11, line 9**: I don't find $Sk_w$ being positive over the desert surprising - it is expected to be positive throughout the CBL. It may be surprising because LES typically give negative values of $Sk_w$ near the surface, which is an artificial result usually due to SGS motions. What is surprising is the difference in sign between the forest and desert, but you've already attributed that to the presence of the canopy;

- **page 11, line 10**: State here as well that the increase in $Sk_w$ is likely due to the height change;

- **page 11, line 10**: 23rd instead of 23th;

- **page 12, line 9**: 23rd instead of 23th;

- **page 13, line 4**: 23rd instead of 23th;

- **page 13, line 23**: This last part is a remnant from the previous version - it should also be removed;

- **Appendix C**: Usually it is considered enough to simply cite the relevant literature when mentioning how the coordinate systems were rotated. No need to completely describe the

workings of planar fit, since it just lenghtens the manuscript unnecessarily;

**References**

1. Hogan, Robin J., et al. "Vertical velocity variance and skewness in clear and cloud-topped boundary layers as revealed by Doppler lidar." Quarterly Journal of the Royal Meteorological Society 135.640 (2009): 635-643.

---

## Author Response (AR2)

**Response to the Reviewer.**

**In most cases, the responses of the authors have been acceptable. Specifically with the raising of the mast over the desert on the 23rd, most of the comments and analyses have been modified to reflect this more objectively and correctly. Some of the other weaknesses have also been appropriately addressed, e.g. justifying the constant flux layer hypothesis. The authors also interpret results more cautiously when attributing the observed differences in the imbalance term between the forest and the desert, to a secondary circulation.**

**However, the manuscript still feels incomplete, particularly because of the heavily underutilized Doppler wind lidars. As it currently stands, these lidars are only mentioned in Appendix B - which is not even referenced in the main text at all. As such, Figure 7 contradicts the time series of <W> from the sonics - but this is not explained or even mentioned at all in the present version of the manuscript.**

**I propose another major revision of the manuscript, by incorporating more analyses involving the two Doppler wind lidars. In doing so, the authors have a unique opportunity to provide another view of the secondary circulations which hasn't been explored so far. I believe that such analyses will substantially increase the value of this study.**

We thank the reviewer for the constructive comments and suggestions.

**Major comments**

**Page 4, line 20: Although you apply planar fit now, you still don't incorporate the directional shear v'w', in the definition of the friction velocity. It's likely it won't alter the current u\* values that much, but it should still be included for completeness;**

It is included in the computations. The text is updated to reflect this.

**page 5, line 28: how was $\delta$ obtained?**

The boundary-layer was detected from a decrease in the backscatter coefficient between the aerosol-laden boundary layer and the clean free atmosphere following the method outline in Münkel et al. (2007). However, the boundary-layer height is not used in manuscript anymore and we didn't include this to the methods for this reason. We have now removed the mention of the boundary layer height since it is not relevant to the discussion anymore.

Münkel C., Eresmaa N., Räsänen J., Karppinen A. (2007): Retrieval of mixing height and dust concentration with lidar ceilometer. Boundary-Layer Meteorology 124(1):117-128

**page 7, line 1: Is the mean vertical velocity that is plotted in Fig. 2, the vertical velocity that is rotated with the planar fit procedure?**

It is the rotated vertical velocity. The text is updated to reflect this.

**Fig. 2 and Fig. 7: This is where the major drawback of this study lies: In addition to these two figures contradicting each other, it remains unclear why that would be so since the authors do not address it at**

**all. First, the authors do not compare Fig. 7 with Fig. 4a,b in Eder et al (2015, AFM). Their LES results and your lidar results are in agreement (at least in terms of the sign of average vertical velocity).**

The results of Fig. 7 are in qualitative agreement with Fig. 4 of Eder et al. (2015), but it should be that noted that their Fig. 4 shows a spatial and temporal average for a 30 min interval, while our results are a temporal average over multiple days. Because of this discussion, we realize that figure 2 and 7 should not be compared with each other as they represent different scales in space and time. Figure 2 shows the nature of fine scale turbulence, while figure 7 shows more of the large scale motions spanning the whole boundary layer. However, we have updated figure 7 with the figure attached in this letter.

**Second, it's obvious from Fig. 7 that at the lowest range gates of both lidars, the vertical velocities converge to zero, perhaps maybe even changing signs very near the surface - which would then be in accordance with Fig. 2. However, this area is under the influence of the dead zone from the lidars, so only speculation is allowed. Even so, it remains clear from Fig. 7 that, throughout the CBL, it is the desert that experiences a downdraft and the forest is the one associated with an updraft. This is in contradiction with the general conclusion of the study, i.e. with the orientation of such a secondary circulation that the authors suggest exists (based on Imb and triple-order moments analyses).**

We want to structure our reply in four parts: first, we discuss the quality of $w$ from the Doppler lidar, then further indicators for a change of sign in $w$ near the surface are discussed and, lastly, our thinking about the contradiction.

(1) The $w$ from the Doppler lidars is uncorrelated with the horizontal wind speed and direction, which indicates that the measured $w$ is not a projection of the horizontal wind speed into the vertical component due to bad instrument leveling. Further, the measured profiles agree with results of highly detailed LES simulations of Yatir forest by Kröniger et al. (accepted and in production).

Kröniger, K., De Roo, F., Brugger, P., Huq, S., Banerjee, T., Zinsser, J., Rotenberg, E., Yakir, D., Rohatyn, S., and Mauder, M. (2018). Effect of secondary circulations on surface–atmosphere exchange of energy at an isolated semi-arid forest. Boundary-Layer Meteorol., Accepted, doi: 10.1007/s10546-018-0370-6.

(2) Averaged $w$-profiles from the Doppler lidars for different times of the day show that during the night the desert has larger $w$ compared to the forest up to 500 m a.g.l. But this phenomenon sinks towards the surface during the day and was partly masked in previous version of Fig. 7 of the manuscript by the height averaging bins of $z/\delta$. Figure 1 below shows separate $w$-profiles for different times of the day without normalization by $\delta$, where the described phenomena can be seen. As the reviewer has mentioned, the profiles above the desert and forest do actually cross over, as seen from figure 1 below. Especially during noon time, the forest is associated with downdrafts and the desert is associated with updrafts. From figure 1, noon and daytime profiles show this cross over at the lowest levels. This phenomenon is observed from the lidars up to a height of about 100m. So we have updated figure 7 with this figure.

[Figure]

Figure 1: Vertical mean velocity profile averaged from $18 - 29$ August (only times with both instruments simultaneously online and the nearest three range gates are discarded). Left to right: Four hour window centered on noon (this corresponds to the old Fig. 7 of the manuscript), day time (sunrise to sundown) and night time (sundown to sunrise). The forest is shown as a solid red line, the desert as solid black line, and a vertical line at $\overline{w} = 0$ as black dashed line. Note that near the surface the desert has always larger w, but only during the noontime with the updrafts of the forest is a change of sign.

(3) We have yet to understand this phenomenon, but we believe it is connected to the particular topographical surroundings of the measurement sites. This educated guess is supported by positive w near the surface in LES of Yatir forest without topography (Eder et al., 2015 or Kröniger et al., accepted and in production). Also in terms of foot-prints, the w of the EC station only account for the very near surrounding of the measurement sites, while mid boundary-layer measurements of the Doppler lidars are more representative of the forest as a whole.

Another insight from this discussion is that the secondary circulation cannot be thought of as a single large rotational system spanning from the desert and forest, rather a much more complex and three dimensional structure. *Close to the surface layer and the canopy sublayer, the transport of energy is indeed from the desert to the forest (Fig. 5 in the manuscript draft). Further, we observe that the desert has more updrafts and the forest has more downdrafts close to the surface (Fig. 1 and 7 (new) in the manuscript draft). However, as we go up above roughly 100m, this behavior flips (new Fig. 7 in the manuscript draft). Lastly, Kröniger et al. (accepted and in production) found in his simulations that large rotational systems developed at specific locations connected to surface features. Therefore, we conclude that the bulk transport in the convective mixed layer by a secondary circulation is from the forest to the desert, but advected with the mean wind and heavily influenced by surface features on a smaller scale than the forest itself. This is highlighted more in the text.*

**This raises another question: does the secondary circulation extend all the way to the top of the CBL or just to a certain z/delta The discrepancy explained in the previous paragraph would suggest the latter. As pointed out by the authors, an answer may lie in the vertical velocity skewness Skw. A simple analysis to do would be to, for available times that the lidars were working, obtain vertical profiles of Skw and inspect their temporal evolution over both the desert and the forest. E.g. plotting a time-height Howmoller diagram for the whole 12 days of the campaign comes to mind. Since the forest experiences negative Skw, it would be interesting to see how far up this persists. Of course, because a third-order**

**moment is involved, a longer averaging period is required - for instance 1 (maybe even 2) hour (thus skipping the single VAD scan that was performed in between vertical stares).**

We believe the secondary-circulation does extend all the way up the boundary-layer top based on two results from other studies (below a short summary):
1) Brugger et al. (2018) observed that the boundary-layer height can be increased by Yatir, if not suppressed by very stable stratification of the free atmosphere or high wind speeds in the boundary-layer. This requires, that the forests effects have already reached the boundary-layer top at the forest site and was in line with an assumed rising of the forest's effects with the convective velocity scale.
2) LES simulations of Yatir forest by Kröniger et al. (accepted and in production) showed the horizontal circulations spanning the atmospheric boundary-layer.

The proposed approach with time-height series of the vertical velocity skewness (shown in Fig. 2 below) was not followed, because the skewness is typically positive throughout a convective mixed layer with narrow, intense updrafts and broader, weaker downdrafts (Hogan et al., 2009). The skewness differences due to turbulence production of a canopy are restricted to the canopy sublayer and therefore not visible in the skewness profile of the Doppler lidar at the forest.

Kröniger, K., De Roo, F., Brugger, P., Huq, S., Banerjee, T., Zinsser, J., Rotenberg, E., Yakir, D., Rohatyn, S., and Mauder, M. (2018). Effect of secondary circulations on surface–atmosphere exchange of energy at an isolated semi-arid forest. Boundary-Layer Meteorol., Accepted, doi: 10.1007/s10546-018-0370-6.

Brugger, P., Banerjee, T., De Roo, F., Kröniger, K., Qubaja, R., Rohatyn, S., Rotenberg, E., Tatarinov, F., Yakir, D., Yang, F. and Mauder, M. (2018): Effect of surface heterogeneity on the boundary-layer height: a case study at a semi-arid forest. Boundary-Layer Meteorology. DOI: https://doi.org/10.1007/s10546-018-0371-5.

Hogan R.J., Grant A. L. M., Illingworth, A. J., Pearson G. N. and O'Connor E. J. (2009): Vertical velocity variance and skewness in clear and cloud-topped boundary layers as revealed by Doppler lidar. *Q. J. R. Meteorol. Soc.* 135: 635–643 (2009), DOI: 10.1002/qj.413

[Figure]

*Figure 2: Time-height series of the vertical velocity skewness for the forest (top) and desert (bottom) from the Doppler lidar measurements.*

**Another useful avenue that would go along the 'eddy size distribution' line-of-thinking that the authors mention in the beginning of the Conclusion, would be to spectrally decompose the lidar-derived Skw. This has been done on several occasions (Hogan et al, 2009, their Fig. 7). By doing this for each range gate, a height-frequency Howmoller diagram could be designed (for a single representative case study, not all 12 days). Such a diagram would then elucidate how each scale of motion contributes to overall Skw. I'm certain some interesting differences between the forest and the desert would become apparent.**

As noted above, the LIDAR data can show the skewness only for the convective mixed layer so we are not proceeding forward with this analysis. However, our team is conducting detailed LES studies of the forest-desert system and we will incorporate this analysis with the LES data in a future publication.

**page 11, line 6: I'm still not comfortable with bringing ejections and sweeps into the mix here (as I've pointed out in my first response). I agree that they are contributors to third-order moments, but the way you describe it currently makes it seem like ejections and sweeps are related to secondary circulations - which is not true. A word of caution would help here, disclosing the fact that ejections/sweeps are a much finer-scale, smaller-scale motions than secondary circulations.**

We have added a note mentioning that all structures visible here are due to the impact of fine scale turbulence.

**In line with the spectral decomposition of lidar-derived Skw I mentioned earlier, I wonder if decomposing the mixed third-order moments would reveal some aspects at the low-frequency part of those cospectra? Significant deviations should be expected there, considering the different ranges of signs (between desert and forest) that you obtained in Fig. 6.**

Again, this study will be conducted in future with detailed LES analysis.

**Also, the fact that higher-order moments require longer averaging times than means and variances, makes me question how reliable the different signs between the forests and desert actually are. Sensitivity analyses of <w'w'u'>, <w'w'w'> and <w'w'T'> as a function of increasing the averaging time with increments of 30 minutes, would make the results more bulletproof.**

Since all the statistics are reported for 30 min periods, we have retained the 30 min averaging, in order to study and compare the same dynamics from different statistics. Changing the length and time scale of the observation, as observed from the difference of LIDAR/EC data in terms of the circulation might tell us about a different dynamics which we might not be studying.

**Minor Comments**

**page 2, line 5: having both or and etc. is redundant, keep just one of them;**

Corrected

**page 4, line 16: it's usually referenced as Monin-Obukhov Similarity Theory (instead of Stability);**

Corrected

**page 8, line 7: 23rd instead of 23th;**

Corrected

**page 8, line 12: 23rd instead of 23th;**

Corrected

**page 9, line 1: in the constant flux layer;**

Corrected

**page 9, line 12: replace summation with sum**

Corrected

**page 10, line 3: 23rd instead of 23th;**

Corrected

**page 10, line 12: 23rd instead of 23th;**

Corrected

**page 11, line 9: I don't find Skw being positive over the desert surprising - it is expected to be positive throughout the CBL. It may be surprising because LES typically give negative values of Skw near the surface, which is an artificial result usually due to SGS motions. What is surprising is the difference in sign between the forest and desert, but you've already attributed that to the presence of the canopy;**

Noted now in the text

**page 11, line 10: State here as well that the increase in Skw is likely due to the height change;**

Stated

**page 11, line 10: 23rd instead of 23th;**

Corrected

**page 12, line 9: 23rd instead of 23th;**

Corrected

**page 13, line 4: 23rd instead of 23th;**

Corrected

**page 13, line 23: This last part is a remnant from the previous version - it should also be removed;**

Corrected

**Appendix C: Usually it is considered enough to simply cite the relevant literature when mentioning how the coordinate systems were rotated. No need to completely describe the workings of planar fit, since it just lengthens the manuscript unnecessarily;**

Just mentioned the reference now.

[revised manuscript text omitted]

**Appendix C:**

~~A planar fit method described by Wilczak et al. (2001) is used, which considers the tilt of the sonic anemometer with respect to local streamlines. Hence this technique is deemed to be more suitable in the presence of sloped terrain, which can be associated with a net non-zero mean vertical velocity. The hypothesis being employed is that under the condition of tilting relative to the local streamlines, a portion of the horizontal velocities will be present in the measured vertical velocity component:~~

$$\overline{w}_m = b_1 \overline{u}_m + b_2 \overline{v}_m,$$

~~where over-bars denote half hour time averaging and the subscript $m$ indicates measured velocity components. Note that this is a modification of the original Wilczak et al. (2001) formulation by Van Dijk et al. (2004), who noted that the equation **??** does not need an additional constant. $b_1$ and $b_2$ are computed using a bilinear regression using all the data points for individual sonic anemometers. Next, to orient the $z$ axis perpendicular to the local streamlines, a rotation is performed:~~

$$[u_{pf}; v_{pf}; w_{pf}] = [M_{pf}][u_m; v_m; w_m]$$

where the rotation matrix is defined as

$$M_{pf} = \begin{bmatrix} cos\alpha & 0 & -sin\alpha \\ 0 & 1 & 0 \\ sin\alpha & 0 & cos\alpha \end{bmatrix} \begin{bmatrix} 1 & 0 & 0 \\ 0 & cos\beta & sin\beta \\ 0 & -sin\beta & cos\beta \end{bmatrix},$$

where $tan\,\alpha = -b_1$ and $tan\,\beta = b_2$ (from which $sin\,\alpha$, $cos\,\alpha$, $sin\,\beta$ and $cos\,\beta$ can be computed). Finally, another rotation is applied to align the wind vector secondary circulation cannot be thought of as a single large rotational system spanning from

5   the desert and forest, rather a much more complex and three dimensional structure. Close to the surface layer and the canopy sublayer, the transport of energy is indeed from the desert to the forest (Fig. 5). Further, we observe that the desert has more updrafts and the forest has more downdrafts close to the surface. However, as we go up above roughly 100m, this behavior flips. Lastly, Kroeniger et al. (2018) found in his simulations that large rotational systems developed at specific locations connected to surface features. Therefore, we conclude that the bulk transport in the convective mixed layer by a secondary circulation is

10   from the forest to the desert, but advected with the mean wind direction:

$$u_{rot} = u_{pf} \cos\theta + v_{pf} \sin\theta; v_{rot} = -u_{pf} \sin\theta + v_{pf} \cos\theta; v_{rot} = w_{pf},$$

where $\theta = tan^{-1}(v_{pf}/u_{pf})$. This also ensures that the crosswind ($v$) component is zero 
[revised manuscript text omitted]
 Dijk, A., Moene, A., De Bruin, H., et al.: The principles of surface flux physics: theory, practice and description of the ECPACK library, Internal Rep, 1, 99, 2004.

van Heerwaarden, C. C. and Guerau de Arellano, J. V.: Relative humidity as an indicator for cloud formation over heterogeneous land surfaces, Journal of the Atmospheric Sciences, 65, 3263–3277, 2008.

Van Heerwaarden, C. C., Mellado, J. P., and De Lozar, A.: Scaling laws for the heterogeneously heated free convective boundary layer,
20  Journal of the Atmospheric Sciences, 71, 3975–4000, 2014.

Wilczak, J. M., Oncley, S. P., and Stage, S. A.: Sonic anemometer tilt correction algorithms, Boundary-Layer Meteorology, 99, 127–150, 2001.

Yang, B., Raupach, M. R., Shaw, R. H., Tha, K., Paw, U., and Morse, A. P.: Large-Eddy Simulation of Turbulent Flow across a Forest Edge. Part I: Flow Statistics, Boundary-Layer Meteorology, 120, 377–412, 2006.

25  Yue, P., Zhang, Q., Wang, R., Li, Y., and Wang, S.: Turbulence intensity and turbulent kinetic energy parameters over a heterogeneous terrain of Loess Plateau, Advances in Atmospheric Sciences, 32, 1291–1302, 2015.

Zhuang, Y. and Amiro, B.: Pressure fluctuations during coherent motions and their effects on the budgets of turbulent kinetic energy and momentum flux within a forest canopy, Journal of Applied Meteorology, 33, 704–711, 1994.